# Variation and prognostic potential of the gut antibiotic resistome in the FINRISK 2002 cohort

Katariina Pärnänen [1] ✉, Matti O. Ruuskanen[1,2], Guilhem Sommeria-Klein[1,3], Ville Laitinen[1], Pyry Kantanen[1], Guillaume Méric [4,5,6,7], Camila Gazolla Volpiano[4,5], Michael Inouye[4,5,8,9], Rob Knight[10], Veikko Salomaa [2], Aki S. Havulinna [1,2,11], Teemu Niiranen [2,12] & Leo Lahti [1] ✉

The spread of antibiotic-resistant bacteria has severely reduced the efficacy of antibiotics and now contributes to 1 million deaths annually. The gut microbiome is a major reservoir of antibiotic resistance in humans, yet the extent to which gut antibiotic resistance gene load varies within human populations and the drivers that contribute most to this variation remain unclear. Here, we demonstrate, in a representative cohort of 7095 Finnish adults, that socio-demographic factors, lifestyle, and gut microbial community composition shape resistance gene selection and transmission processes. Resistance was linked not only to prior use of antibiotics, as anticipated, but also to frequent consumption of fresh vegetables and poultry, two food groups previously reported to contain antibiotic-resistant bacteria. Interestingly, resistance was not linked to the consumption of high-fat and high-sugar foods, but was consistently higher in females and urban high-income individuals, who currently have generally lower mortality rates. Nevertheless, during the 17-year follow-up, high resistance was associated with a 1.07-fold increase in mortality risk, comparable to elevated blood pressure, and with a heightened risk of sepsis. These findings highlight risks and socio-demographic dimensions of antibiotic resistance that are particularly relevant in the current context of global urbanization and middle-class growth.

Antimicrobial resistance, and antibiotic-resistant bacteria (ARB) in particular, represent a growing threat to global health, projected to cause 10 million deaths annually by 2050, surpassing fatalities from any other diseases[1]. In bacteria, the resistance is encoded by antibiotic resistance genes (ARGs), for which the gut serves as a major reservoir in humans[2,3]. The occurrence of ARGs in gut bacterial communities is driven by the ecological processes of transmission and selection (Fig. 1)[2,4,5]. Transmission of resistant bacteria from other individuals,

[1]Department of Computing, University of Turku, Turku, Finland. [2]Finnish Institute for Health and Welfare (THL), Helsinki, Finland. [3]Inria, Univ. Bordeaux, INRAE, Talence, France. [4]Cambridge Baker Systems Genomics Initiative, Baker Heart and Diabetes Institute, Melbourne, VIC, Australia. [5]Department of Cardiometabolic Health, University of Melbourne, Melbourne, VIC, Australia. [6]Central Clinical School, Monash University, Melbourne, VIC, Australia. [7]Department of Cardiovascular Research, Translation and Implementation, La Trobe University, Melbourne, VIC, Australia. [8]Cambridge Baker Systems Genomics Initiative, Department of Public Health & Primary Care, University of Cambridge, Cambridge, UK. [9]BHF Cardiovascular Epidemiology Unit, Department of Public Health & Primary Care, University of Cambridge, Cambridge, UK. [10]University of California San Diego, San Diego, CA, USA. [11]Institute for Molecular Medicine Finland, FIMM-HiLIFE, Helsinki, Finland. [12]Department of Internal Medicine, Turku University Hospital and University of Turku, Turku, Finland. ✉e-mail: katariina.parnanen@utu.fi; leo.lahti@utu.fi

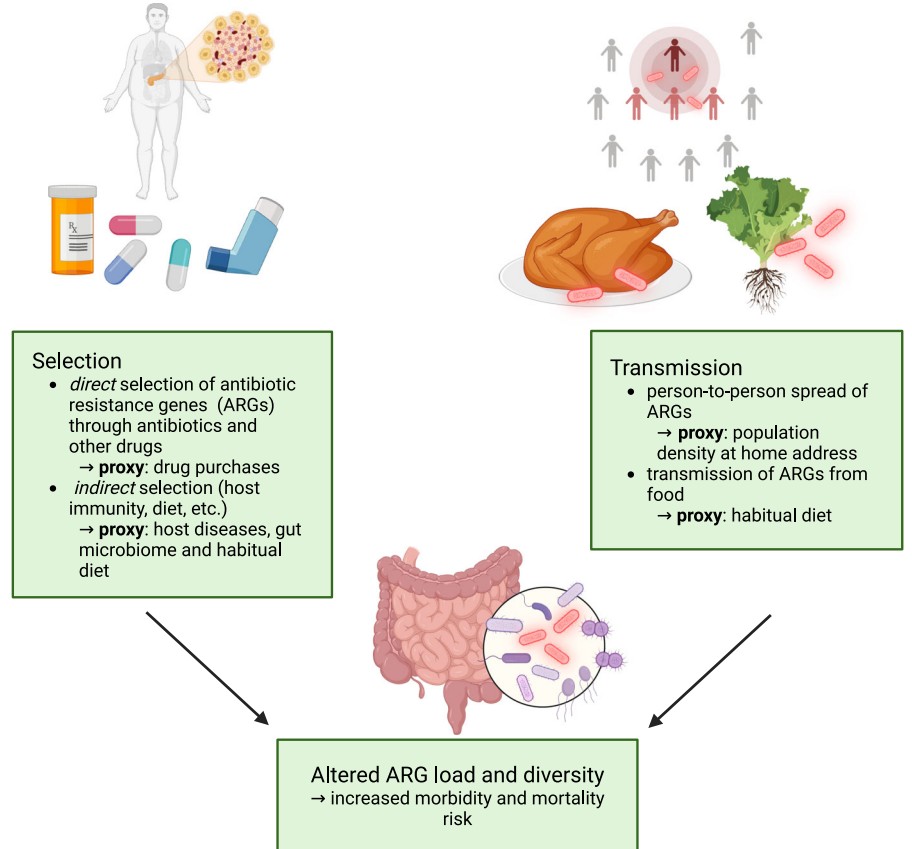

Fig. 1 | **Ecological and epidemiological framework for population-scale variation in resistome composition.** Key ecological processes underlying antimicrobial resistance gene (ARG) variation include (1) *selection* of antimicrobial resistance genes and bacteria that carry them, for instance as a *direct* consequence of antibiotic consumption or *indirectly* due to environmental selection or host factors (e.g., prevalent disease) that impact the growth of those bacteria; (2) *external transmission* via antimicrobial resistance gene carrying bacteria. We associated resistome diversity, composition, and the overall resistance gene load with socio-demographic parameters - some of which can be used as proxies for the above ecological processes - as well as long-term morbidity and mortality risk. Created in BioRender https://BioRender.com/p8092rg.

food, or the environment is an established driver of resistance[4,6,7]. Direct selection of ARGs is caused by antibiotic use, which selects resistant strains from the local pool, either acquired through transmission or locally evolved. Finally, indirect selection of resistant bacteria is caused by environmental factors that favor the growth of bacterial taxa prone to harbor ARGs, for instance, due to previous frequent exposure to antibiotics or high rates of horizontal gene transfer[8,9]. Despite the far-reaching public health implications, there is limited research on the factors that influence the human gut resistome−i.e., the prevalence and load of different ARGs in the human gut microbiome−in contrast to the extensive research on cancer and cardiovascular disease.

In order to tackle this crisis, the World Health Organization has called for research on the demographic parameters underlying the emergence and spread of antimicrobial resistance in human populations[10]. Previous studies based on country-level statistics have shown that resistance varies between countries[6,11,12] and is influenced by national antibiotic use patterns[13,14]. However, few studies have analyzed this variation using large population cohorts, which are crucial for identifying socio-demographic risk factors for resistance beyond nationality and antibiotic use, investigating the underlying ecological and epidemiological processes, and assessing the risks associated with higher resistance.

In this study, we analyzed fecal samples collected in 2002 in a representative population cohort of 7095 adults (mean age 49, 55% women) without acute infections, from six contrasted Finnish regions (Fig. 2a; Supplementary Fig. 1; FINRISK[15]). Data on baseline individual address-level geographic location, diet, household income level, prescription drug purchases, diseases, and causes of death up to 2019 were gathered from electronic population registers, health examinations, and questionnaires (see "Methods" section and Supplementary Table 1). We describe how socio-demographic factors and gut microbiome composition drive variations in antibiotic resistance, and we demonstrate that antibiotic resistance predicts increased mortality and morbidity risk on decadal time scales.

## Results

### Socio-demographic drivers of resistance selection and transmission

We assessed the relative importance of different drivers in shaping individual resistome by training a supervised machine learning model (boosted GLM, which includes a variable selection procedure; see "Methods" section) to predict the total ARG load− defined as the total abundance of all ARGs, measured in log10 reads per kilobase per million reads (RPKM) −from demographic, health and lifestyle factors across the population. We also included the relative abundances of major bacterial families in the gut as predictors (Supplementary Fig. 2a), because gut microbiome composition is known to play a role in shaping the resistome[2,8,16]. The model explained 31% of ARG load variation on a set of left-out test participants, and 29% without including microbiome composition (Fig. 2b, Supplementary Fig. 2b). Past antibiotic use emerged as the most important predictor (27% of the variance; linear model including all relevant predictors, see "Methods" section), followed by the relative abundances of major

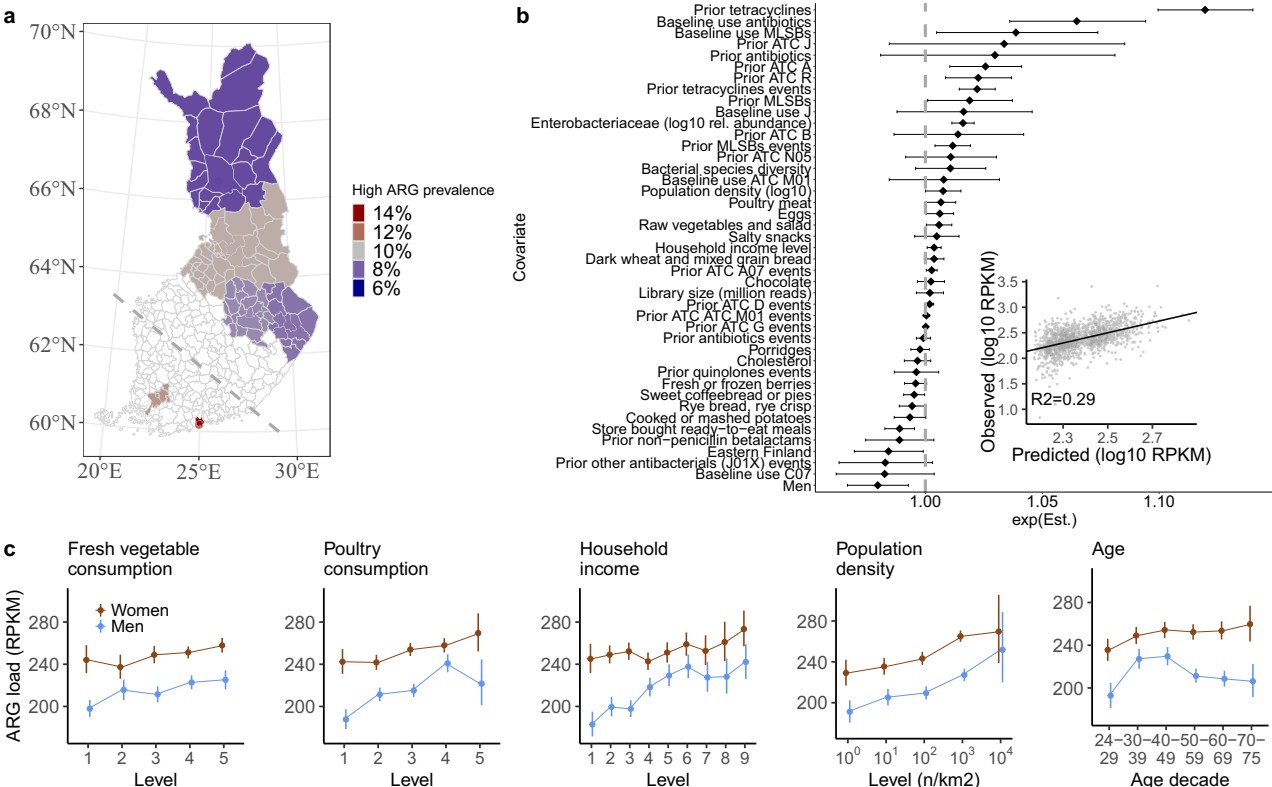

**Fig. 2 | Predictors and variation of antimicrobial resistance gene (ARG) load.** **a** Regional variation in total ARG load across the six regions in the FINRISK cohort ($N$ = 7095 participants, from North to South: Lapland, Oulu region, North Savonia and North Karelia in the East, and Turku and Helsinki urban regions in the West; the dashed line illustrates the East-West split). The color indicates the fraction of the population in the high-ARG load group for each region (the top-10% quantile in this cohort, i.e., participants with a total abundance of all normalized ARGs above 458 reads per kilobase per million reads - RPKM; Supplementary Table 3). **b** Drivers of ARG load ($N$ = 7095 participants, train/test split 70%/30%, 4397 degrees of freedom, boosted GLM; see "Methods" section). For drivers of resistome diversity, see Supplementary Fig. 2a. The line plot shows each predictor's exponentiated estimate exp(Est.) with points and the 95% confidence interval as bars; 1.05 exp(Est.) corresponds to a 5% increase in ARG load per change of unit in the covariate. The inset shows the predicted and observed ARG load in test data ($R^2$ = 0.29). Prior X:

purchase of drug X in the seven years before sampling (yes/no); Prior X events: number of purchases of drug X in the seven years before sampling; Baseline X: Purchase of drug X in the six months before sampling (yes/no). Anatomic therapeutic class (ATC) abbreviations: A = Alimentary tract and metabolism, A07 = Antidiarrheals, intestinal anti-inflammatory/anti-infective (includes antibiotics for intestinal infections), B = Blood and blood forming organs, C07 = Beta blocking agents, D = Dermalogicals, G = Genito-urinary system and sex hormones, agents, J = Anti-infectives, M01 = Anti-inflammatory and antirheumatic products, N05 = Psycholeptics, R = Respiratory system. **c** Association between ARG load and raw vegetables and poultry consumption, household income, population density, and age ($N$ = 7095 participants). The points represent the mean, while the bars indicate the 95% confidence interval: see the "Methods" section for variable descriptions and Supplementary Data 3 for the numerical estimates. Similar trends can be observed within individual regions (Supplementary Fig. 3).

bacterial families (3%), demographic variables (household income and sex; 2%), geography (East/West; 1%) and diet (1%). Similar trends could be observed independently within each of the six geographical regions represented in our population cohort (Supplementary Fig. 3, Supplementary Table 2-S3). In the following, we discuss how these different predictors shed light on the ecological and epidemiological processes underlying ARG selection and transmission (Fig. 1).

## Drug use

We studied the effect of drug consumption in our cohort based on the Finnish digital register for drug purchase reimbursements, which was established seven years before sampling. Antibiotic use causes direct selection of antibiotic resistance genes[17]. We found that ARG load increased in the years following the use of several classes of antibiotic and non-antibiotic therapeutics listed in the Anatomical Therapeutic Chemical Classification System ($N$ = 7095, 70%/30% train/test split, ATC; log-linear model, $P < 0.05$ for several classes, Supplementary Data 1). Purchase of antimicrobials (ATC class J) in the previous seven years was associated with a 55% higher ARG load ($P < 10^{-3}$) compared to individuals who made no antimicrobial purchases in that period. In particular, we detected a 67% increase in ARG load ($P < 10^{-3}$) following

the purchase of tetracycline (ATC class J01A), a widely used antibiotic class in both humans and animal production[18] (second most frequently purchased antibiotic class after penicillin in the cohort, Supplementary Table 1) and a 39% increase following the purchase of macrolide antibiotics (ATC class J01F; $P < 10^{-3}$). Recent purchase of macrolide (i.e., in the 6 months prior to sampling) was associated with a markedly higher increase (68% increase in ARG load, 95% confidence interval (CI) 57%–80%, $P < 10^{-3}$, Supplementary Data 1), which suggests that time since purchase can have an important effect on resistance for some antibiotics. However, when considering all antibiotics combined, there was no significant difference between a recent purchase and a purchase in the previous seven years (Supplementary Data 1, 95% CIs 43%–55% and 49%–60%, respectively). These associations remained robust after controlling for other covariates in the model (boosted GLM, Fig. 2b). Besides antibiotics, purchases of respiratory medication (ATC class R) were also associated with a higher ARG load after controlling for past antibiotic use, demographic factors, and other participant-level data (Fig. 2b). Although this hints at a possible selective effect of these non-antibiotic drugs on resistance, drug use could be a proxy for additional, unobserved covariates, such as hospitalization or antibiotic use prior to our records. However, there was

no significant association between past-year hospital exposure and higher ARG load or diversity after controlling for antibiotic use ($N = 7095$, 70%/30% train/test split, linear model, $P > 0.05$, CIs $-2\% - 6\%$ and $-5\% - 3\%$, respectively). Other associations between non-antibiotic drugs and ARG load were not robust to adjusting for other covariates or microbiome composition at the bacterial family level (boosted GLM, Fig. 2b, Supplementary Fig. 2). Overall, these results confirm the major role of direct selective pressure by antibiotics on resistance load, and highlight the need to consider the long-term impact of antibiotic use.

## Diet

To investigate the role of food in resistance selection and transmission, we associated ARG load with the self-declared habitual consumption frequency of 42 food groups (5 frequency levels; see "Methods" section, Fig. 2b, c, Supplementary Fig. 2)[19]. Poultry had the strongest positive association with ARG load ($N = 7095$, 70%/30% train/test split, log-linear model, 4% average increase per consumption level, adjusted for antibiotic use; $P < 10^{-3}$; Supplementary Data 1), followed by raw vegetables and salad (3% increase per level; $P < 10^{-3}$). These associations remained robust even after controlling for other covariates in the model ($N = 7095$, 70%/30% train/test split, boosted GLM, $P = 0.04$ and 0.03, respectively; Fig. 2b, c) and had similar trends in all regions (Supplementary Fig. 3). Moreover, their association with ARG load remained significant also when controlling for microbiome composition at the bacterial family level ($P = 0.004$ and 0.05, respectively, see "Boosted GLMs" section). This suggests that diet can contribute to resistance independently of its selective effect on microbiome composition, probably through the transmission of ARGs to the gut[20,21]. These ARGs are most likely transmitted as live ARB (in raw vegetables and salad, or poultry contaminated during processing), although extracellular ARGs can also be transmitted from cooked food[22–24]. We did not find a similar association for beef or pork, which is in line with the fact that other meat production animals in Finland were reported to have much less resistance than poultry at the time of sampling[25]. In particular, tetracycline, macrolide, beta-lactam, and amphenicol, which were all common antibiotic resistance classes in the FINRISK cohort (Supplementary Fig. 4), were common in poultry in 2002 but much less so in cattle according to surveillance reports[25].

Food may also contribute to resistance through indirect selection[26], that is, by promoting or inhibiting the growth of taxa that tend to carry more ARGs, such as *Enterobacteriaceae*. In particular, it has been hypothesized that high-fat, high-sugar, and processed foods could promote the growth of such taxa. The relative abundance of many bacterial families in the gut microbiome varied with diet ($N = 7095$, 70%/30% train/test split, linear model, $P < 0.05$, Supplementary Data 2). For example, *Bifidobacteriaceae* were positively associated with yogurt, high-sugar foods, such as candy, and simple carbohydrates, such as pasta; *Prevotellaceae* with red meat, rye bread and porridge; and *Bacteroidaceae* with fast foods, snacks, and ready-to-eat meals. In contrast, *Enterobacteriaceae* were negatively associated with candy, french fries, and ready-to-eat meals, and were not positively associated with any food groups. We observed positive associations with ARG load for only some of these foods, such as chocolate and fast food, and these associations were weak and not robust to adjusting for participants' covariates (log-linear model, 1% increase per level, $P > 0.05$; Fig. 2b; Supplementary Data 1). Interestingly, some of these foods were, in fact, inversely associated with ARG load, as were also total cholesterol and BMI ($P = 0.03$ and 0.002, respectively). This might be explained by the lack of raw vegetables and poultry in the typical diets containing these foods (Supplementary Data 2). Conversely, high-fiber foods have been proposed to select against ARB[27]. Associations with high-fiber foods were mixed: both raw and cooked vegetables showed a positive association with ARG load, whereas berries, porridges, and rye bread exhibited an inverse association ($N = 7095$, 70%/30% train/test split, $P = 0.01$, $P = 0.004$, and

0.01, respectively; Supplementary Data 1). Overall, our findings indicate that transmission from food is the main factor in food-related resistance, whereas evidence for indirect selection through diet is limited, with only some fiber-rich foods potentially decreasing ARG load.

## Sex and income

Next, we investigated the impact of sex and income on ARG load (Fig. 2, Supplementary Fig. 2, Supplementary Data 1). Men's average ARG load was 92% that of women (95% CI 89%–94%, $P = 5 \times 10^{-9}$; log-linear model, Supplementary Data 1), a trend that could be observed independently across all six regions (Supplementary Fig. 3). Women tend to purchase more antibiotics (1.4 more purchases on average in a linear model adjusted for age and household income; 95% CI 1.21–1.62, $P < 1 \times 10^{-3}$) and consume raw vegetables more frequently than men (0.56 increase; 95% CI 0.51–0.62, $P < 1 \times 10^{-3}$). Nevertheless, the ARG load difference between sexes remained significant even after controlling for differences in other covariates, and for other covariates and microbiome composition (boosted GLM, $P = 0.002$ and $P = 0.003$, respectively; Fig. 2b, Supplementary Fig. 2b, Supplementary Table 10). Possible causes for this include sex-specific differences in immunity, as well as in occupation, caretaking, prevalence of urinary tract infections[10], and propensity to seek medical care[28], which can all lead to increased ARB exposure. The covariates most associated with ARG load in women were similar to those in the whole cohort (Supplementary Data 1), with tetracycline use having the strongest effect, followed by respiratory drugs.

ARG load increased with household income, showing an average 2% increase per unit increase in income level (ranging from 1 to 9; Supplementary Data 1, log-linear model, $P < 1 \times 10^{-3}$). This association was robust to adjusting for other covariates and microbiome composition (boosted GLM, $P = 0.01$ and $P = 0.04$, respectively; Fig. 2b, Supplementary Fig. 2b,). This finding contrasts with the generally positive association between higher socio-economic status and health[29]. The increased resistance in this subgroup might be explained by lifestyle factors associated with a higher income, such as international travel[30].

## Geographic variations

Whereas earlier large-scale studies have consistently reported variation in resistance between countries[11,13,31], access to participants' home addresses from national population registers allowed us to study fine-grained geographic variations in the resistome. The urban regions around Helsinki and Turku, respectively the largest and third largest cities in Finland, exhibited the highest regional ARG load and were also enriched in individuals with a high-ARG load (top-10% quantile; >458 RPKM; Fig. 2a, Supplementary Table 3). In contrast, the lowest median ARG load was observed in Lapland, a rural region with a remarkably low population density. Compared to Lapland, the median ARG load was 20% higher in Helsinki and 13% higher in Turku. The increase in the prevalence of high-ARG individuals was even higher (84% and 47%, respectively), suggesting that the moderate increase in overall ARG load in urban regions strongly increases the individual risk of acquiring a high-ARG load, possibly because of higher ARG transmission in densely populated areas[32]. This is further supported by the observation that ARG load also increased more generally with population density (Fig. 2b, c; Supplementary Data 1). This observation remained robust even when controlling for other covariates, such as diet, income, and microbiome composition (Supplementary Fig. 2b). This suggests that increased inter-individual ARB transmission is the primary reason for higher ARG load in regions with higher population density. The densely populated urban areas of Turku and Helsinki also receive more international travelers and immigration[33,34], which contributes to the transmission of ARGs from abroad.

In addition, we found regional variations in ARG load that could not be explained by population density. In particular, Eastern Finns

had a generally lower ARG load (log-linear model Supplementary Data 1), a result that was robust to controlling for other covariates in the model including population density, diet and income class (boosted GLM, *P* = 0.04, Fig. 2b) as well as microbiome composition (*P* = 0.02, see "Accounting for microbiome composition" section). This might be explained by the well-known genetic and lifestyle differences between the populations of Eastern and Western Finland[35]. Finally, besides ARG load variations, we also found significant albeit small differences in resistome composition between our six Finnish regions (Supplementary Table 4, PERMANOVA, *P* = 0.002; explained variance <0.1%).

## The role of gut microbiome in shaping the resistome
Besides ARG load, we found the resistome's ARG composition to vary widely among the participants. ARG classes were indeed highly variable in prevalence across the cohort, with only tetracycline resistance detected in virtually all participants (Fig. 3a; Supplementary Fig. 4). We found that individual resistome diversity, defined as the Shannon diversity of ARG alleles, was higher in urban areas and increased with population density, similarly to ARG load, which hints at the likely role of inter-individual transmission in generating resistome diversity[32,36]. Resistome diversity was also partly explained by bacterial species diversity in the gut microbiome (linear regression against species diversity, $R^2$ = 0.11; with all covariates, see Fig. 3b; Supplementary Fig. 2a), a result that further highlights the role of microbiome composition in shaping the resistome. The association between bacterial diversity and ARG load, on the other hand, remained weak (linear regression against species diversity, $R^2$ = 0.001; with all covariates, see Fig. 2b). Furthermore, after identifying the range of possible bacterial hosts for different ARGs through a BLAST search against the *nt* database[37,38], we found phylogenetic clustering among the hosts of several prevalent ARGs (Fig. 3e, see "Methods" section for more details on ARG host identification). This suggests that not only are some bacterial taxa more prone to harbor ARGs, but there is a phylogenetic signal in the specific ARGs they tend to carry[8]. In contrast, the tetracycline resistance genes covered a broad phylogenetic range of bacterial hosts, which could partly explain their high population prevalence. This aligns with the literature, where tetracycline resistance has been found to be the most common class of resistance in the human gut, carried by multiple taxa[2]. Finally, whereas the often-pathogenic bacterial genera *Escherichia* and *Klebsiella* carried the largest number of unique ARGs, several abundant ARGs were carried by commensal taxa: for instance, beta-lactam resistance gene *cfxA6* was only detected in *Bacteroides*, *Parabacteroides*, *Veillonella*, *Butyricimonas,* and *Prevotella* (BLAST *nr* database[37,38], Fig. 3e). This is in agreement with existing literature, where *Bacteroides* have been observed to carry ARGs and *cfxA6* has frequently been identified in *Prevotella*[39,40]. These results highlight the importance of non-pathogenic, commensal taxa as a reservoir of ARGs in the gut. Nevertheless, our methods only allowed us to investigate common bacterial taxa in the gut and might overlook some less common but important pathogenic ARG carriers.

To further characterize co-variation between broad patterns of microbiome composition and resistance, we identified five sub-communities, or *enterosignatures* (ES), defined using non-negative matrix factorization (NMF)[41]. These represent co-varying bacterial genera (Supplementary Fig. 5), which collectively explained 82% of all genus-level variation. They were respectively dominated by members of *Bacteroides* (ES-Bact), *Firmicutes* (ES-Firm), *Prevotella* (ES-Prev), *Bifidobacterium* (ES-Bifi), and *Escherichia* (ES-Esch). Each ES was associated with a different characteristic resistome profile (Fig. 3c, d, Supplementary Fig. 6, Supplementary Table 5). For instance, ES-Prev was associated with high beta-lactam and low tetracycline resistance gene abundances. High ES-Bact and ES-Esch, and low ES-Prev and ES-Bifi, were associated with an increased ARG load. Intriguingly, this

analysis also revealed non-monotonic relations between enterosignature abundance and ARG load, which suggests potentially complex underlying ecological relations. For instance, the prevalence of high-ARG individuals is reduced for moderate amounts of ES-Bact and increases for both low and high abundances of this enterosignature. In ES-Bifi, high-ARG load is primarily associated with the presence/absence, rather than the abundance of this enterosignature (Fig. 3c). These observations are further supported by similar patterns in bacterial families (Supplementary Fig. 7; Supplementary Data 1). High-ARG loads were associated with high *Bacteroidaceae* and *Enterobacteriaceae* abundance and low *Prevotellaceae* and *Bifidobacteriaceae* abundance (Supplementary Fig. 7, Supplementary Data 1). The association between *Prevotellaceae* and low ARG load, in particular, may mediate our finding that some *Prevotellaceae*-associated fiber-rich foods (such as porridge and rye bread) decrease ARG load, as reported in the "Diet" section. Overall, our findings emphasize the role of microbiome composition as a key determinant and a potential mediator of population-level resistome variation.

## Resistance predicts long-term mortality and sepsis risk
Antimicrobial resistance has been linked to increased risks associated with infectious diseases, but its long-term health implications have remained largely uncharacterized in the absence of population studies with individual-level health information and sufficiently long follow-up times. In order to examine the prognostic potential of the gut resistome, we gathered follow-up data from the national health registers on all major health events, including deaths, for all participants from the 2002 baseline sample collection until 2019. In total, 863 (12.2%) of the participants died during the 17-year follow-up period, and 197 (2.8%) of them developed sepsis. We estimated time-to-event associations for selected variables with a probabilistic Cox model (see "Methods" section). We controlled mortality-associated covariates (age, smoking, sex, diabetes, use of antineoplastic and immunomodulating agents, systolic blood pressure, and self-reported antihypertensive medication) and (log10) relative abundance of *Enterobacteriaceae*, a bacterial family known to harbor ARGs[42] and linked to higher mortality[43]. In addition, we controlled for income and raw vegetable and salad consumption as potential confounders, since they have been reported to be inversely associated with mortality[29,44].

We observed a positive association between ARG load (log10 RPKM) and increased mortality risk (probabilistic Cox model, posterior mean Hazard Ratio (HR) 1.34; Fig. 4a, Supplementary Table 6). This change in mortality risk was similar to that induced by systolic blood pressure (mean HR after scaling by standard deviation for comparability: 1.07, 95% CI 1.02–1.13 for ARG load; 1.09, 95% CI 1.03–1.15 for systolic blood pressure; see "Methods" section). The mortality associated with high-ARG load was more significant in women (Fig. 4b).

ARG load was specifically associated with increased mortality due to respiratory causes (ICD-10 R codes; median HR 2.22; Supplementary Fig. 8, Supplementary Table 7), although the sample sizes for cause-specific events, including infectious mortality other than respiratory infections (ICD-10 A codes; *N* = 13), remain relatively low. We also observed an association between ARG load and increased sepsis risk (median HR 2.22; Supplementary Fig. 8, Supplementary Table 8). The associations between ARG load, all-cause mortality, mortality by respiratory causes, and sepsis were significant also when controlling for the abundance of each prevalent bacterial family as a covariate (probabilistic Cox model <5% probability of no association). These observations demonstrate that ARG load represents a robust risk factor for incident mortality and sepsis (Fig. 4; Supplementary Fig. 8).

## Discussion
Our study unveils population variation and prognostic potential of the adult gut resistome in a single, representative population cohort. This represents, to our knowledge, the largest analysis of antimicrobial

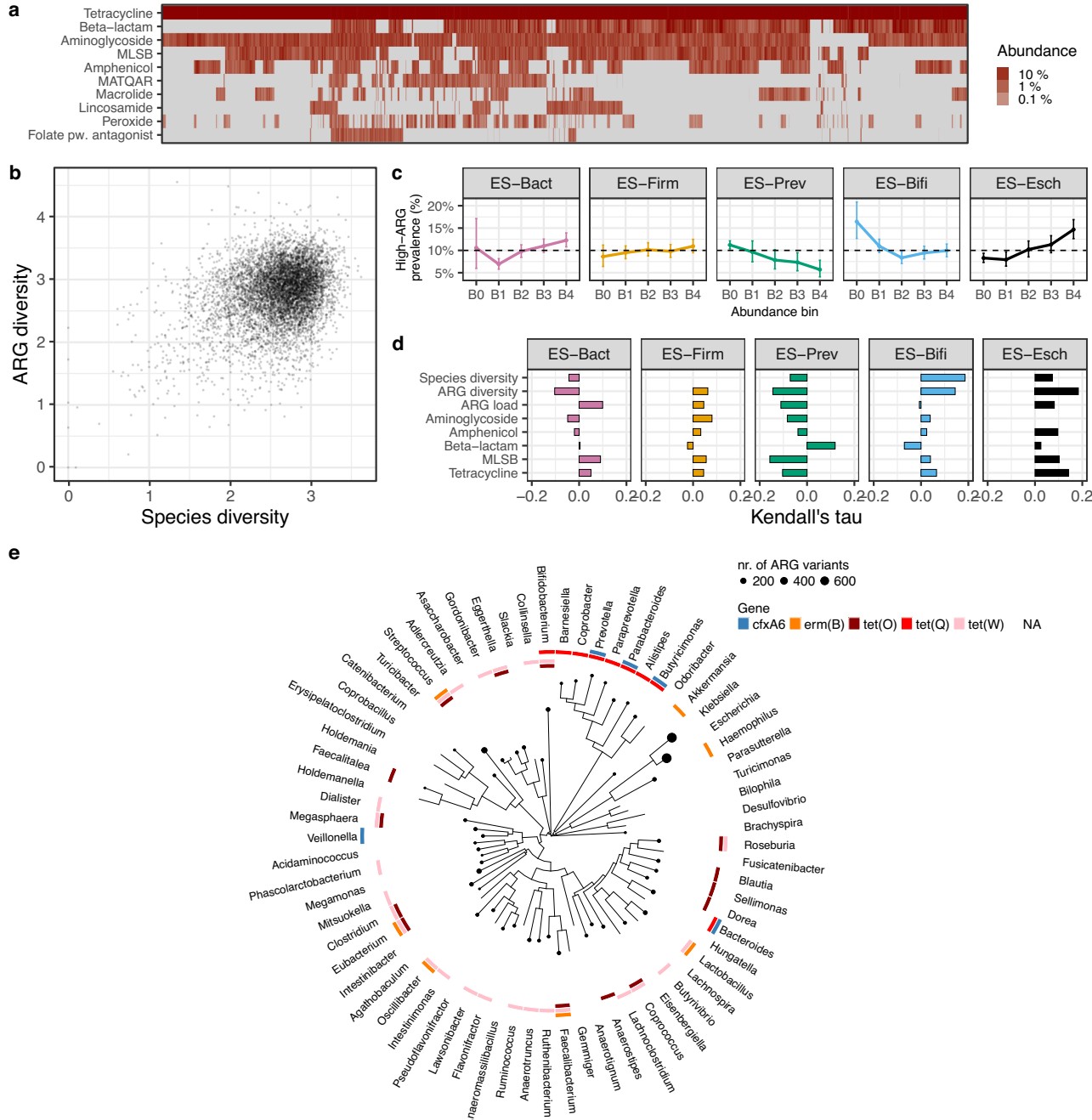

**Fig. 3 | Resistome variation and taxonomic composition. a** Resistome composition: relative abundance of the 10 most abundant ARG classes in the FINRISK cohort (*N* = 7095 participants). **b** Bacterial species diversity and resistome (ARG) diversity (Shannon index; Pearson, *P* < 10⁻³, *r* = 0.32, 95% confidence interval, 0.30–0.34, *N* = 7095 participants). **c** Enrichment of high-ARG load participants (>458 RPKM) as a function of enterosignature abundance (ES) (B0: not detected; B1–B4: 25% abundance quartiles, *N* = 7095 participants); each enterosignature (ES) represents a sub-community of co-varying bacterial genera (Supplementary Fig. 5). ES-Bact is characterized by *Bacteroides*, ES-Firm by Firmicutes, ES-Prev by *Prevotella*, ES-Bifi by Bifidobacteria, ES-Esch by *Escheria*. The estimated prevalence of high-ARG individuals within each abundance bin is shown as points with 95% credible intervals represented by bars (probabilistic Bernoulli model, see

"Methods" section). The dashed line indicates the expected prevalence of the high-ARG individuals in the entire study population (10%). **d** Associations between the abundance of each ES and bacterial species diversity, resistome diversity, total ARG load, and load of the five most dominant ARG classes (two-sided Kendall's Tau with FDR correction; Supplementary Table 5). All associations are significant (*P* < 0.05), except Beta-lactam in ES-Bact and ARG load in ES-Bifi. Abbreviations: MLSB: "Macrolide, Lincosamide, Streptogramin B"; MATQAR: "Macrolide, Aminoglycoside, Tetracycline, Quinolone, Amphenicol, Rifamycin". **e** Phylogenetic relatedness among the most prevalent bacterial genera in the FINRISK cohort and their association with the most prevalent ARGs. Node size indicates the total number of ARGs found in each genus according to the BLAST *nr* database; the colors indicate a match between each genus and the ARGs.

resistance and its socio-demographic determinants in a population study. The most influential factor in predicting the resistance gene load was the prior use of antibiotics. This confirms that the direct selection of resistance genes induced by antibiotic consumption is a key mechanism underlying individual resistance levels, with effects

potentially persisting for several years. Moreover, variation in the gut microbiome composition was linked to variation in the composition and load of resistance genes, which suggests that indirect selection of resistant bacteria plays a role in population-level resistome variation, although we found only limited evidence that diet drives it. Finally, we

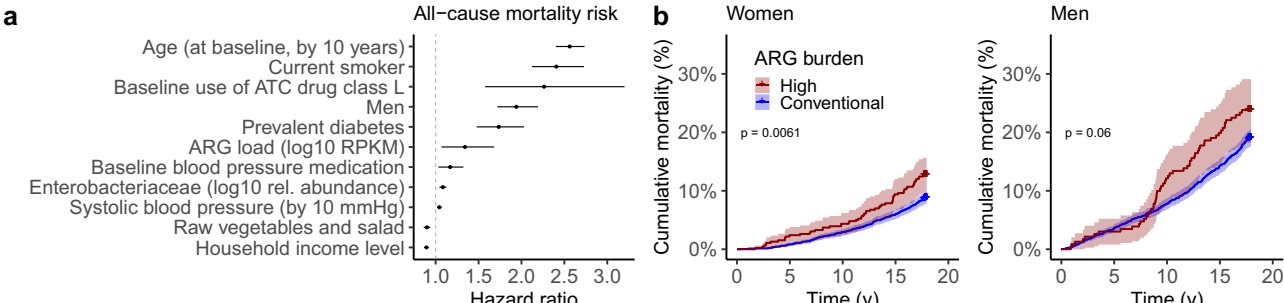

**Fig. 4 | Antibiotic resistance gene load predicts long-term mortality risk. a** Total ARG load predicts long-term mortality risk in a 17-year follow-up (*N* = 7095 participants, probabilistic multivariate Cox proportional hazards model; See Supplementary Table 6). The points indicate the median, while the bars represent the 95% credible interval (CI). The model is adjusted for *Enterobacteriaceae* relative (log10) abundance, age, smoking, sex, diabetes, antineoplastic and immunomodulating agents, body mass index, self-reported antihypertensive medication, systolic blood pressure, recent antibiotics use (six months before baseline), household income, and raw vegetable and salad consumption. The median hazard ratio (HR) is shown per unit increase of each variable, along with the 95% CIs; variables whose CI overlaps with 1 (no association) are excluded from the graph. **b** Cumulative incidence of all-cause mortality during the follow-up period for individuals stratified by high (red; >458 RPKM) and conventional (blue) ARG load. The shaded area represents the 95% CI. High-ARG load is associated with significantly higher mortality among women (*P* = 0.006; log-rank test; multivariate Cox). A similar but non-significant trend is observed in men (*P* = 0.06). Associations of ARG load with sepsis and cause-specific mortality are shown in Supplementary Fig. 8. Here, in a representative cohort of 7095 Finnish adults, the authors reveal that gut antibiotic resistance is shaped not only by antibiotic use but also by the microbiome, diet, lifestyle, household income—and is linked to higher long-term mortality.

found that factors associated with ARG transmission–high population density, high income, and certain food groups–could further improve predictions of individual resistance levels. These findings shed light on the ecological and epidemiological processes that can foster the emergence and spread of resistance genes in human populations. Remarkably, our data suggests that the total abundance of ARGs can serve as a predictor for long-term sepsis risk, and that its effect on mortality risk is comparable to elevated blood pressure over a 17-year follow-up. Accordingly, resistome composition should be included in the ongoing efforts to define a healthy microbiome[45].

Each study must be evaluated within the context of its limitations. Sequence-based identification of ARGs does not ensure phenotypic antibiotic resistance in bacteria; rather, it acts as a proxy for the potential of antibiotic resistance in a microbial community. Moreover, our analysis depends on shallow, short-read metagenomic profiling. This technique emphasizes the most abundant and prevalent ARGs but does not allow for the assembly or creation of draft genomes, which restricts available information about gene variants, their bacterial hosts, and the genomic context of the ARGs. The limited sequencing depth is offset by the sample size, which provides sufficient statistical power for making generalizable predictions and differentiating between the effects of various potential confounders.

The patterns observed in one population may not be applicable to other countries and regions, especially since the Finnish population is relatively homogeneous. However, this homogeneity reduces confounding effects related to genetic diversity, cultural variation, and major differences in healthcare infrastructure, enabling a more accurate assessment of how other covariates–such as antibiotic use, diet, or the microbiome–relate to ARG load and the resistome. This study can serve as a foundation for comparisons with more heterogeneous populations to examine generalizability and broader global trends. Finland has also experienced demographic and agricultural changes in recent decades, which may impact the relevance of the results today. Nevertheless, even if some of the associations we report change with time or between populations, the broader determinants of ARG load– such as antibiotic use, microbiome composition, diet, and socio-demographics–are likely to remain relevant.

The FINRISK study provides the key benefit of long-term follow-up for a well-characterized and highly structured cohort. The lack of longitudinal metagenome data, however, limits our ability to assess causal relationships. Indeed, population-level resistome patterns are expected to evolve over time. This could influence the validity of our mortality predictions, although individuals with high-ARG load at the time of sampling likely remained at elevated levels over the follow-up period. Antibiotic resistance has also been shown to be an indicator of poor overall health status[46], which could bias our predictions. Further research is needed to examine long-term resistome dynamics and their impact on mortality.

Our findings highlight antimicrobial resistance as a potential disease of affluence, with higher prevalence among high-income individuals, women, and urban populations. Yet, most antimicrobial resistance-related mortality currently occurs in low- and middle-income countries, infants, and the elderly[47]. Antibiotic resistance-related deaths are rare in Finland; only three deaths were estimated to be caused by antibiotic-resistant bacteria in 2002, according to the European Centre for Disease Prevention and Control[48]. Although the contribution of antibiotic resistance to deaths is likely under-reported, antibiotic resistance-related mortality is much lower than that of cancer and cardiovascular disease, which are more prevalent in rural, low-income populations and men[28,29]. Therefore, despite our finding of an increased mortality risk with higher ARG load, overall mortality remains lower among women, high-income individuals, and urban populations. However, the increase in resistance-related deaths over time[47,49] might elevate the relative mortality risk among these demographics. Indeed, although antibiotic use in Finland has declined since 2002, global antibiotic consumption and resistance-related mortality continue to rise[47,50,51]. Current trends in lifestyle, including increased urbanization and demand for poultry, might further exacerbate antimicrobial resistance-related mortality globally[36,51,52], as well as in Finland. Our findings serve as early indicators of the growing significance of antimicrobial resistance as a contributor to overall mortality[47,53].

## Methods
### Study participant details
The FINRISK population surveys were conducted every five years from 1972 to 2012 with the primary objective of tracking trends in cardiovascular disease risk factors in the Finnish adult population. The FINRISK 2002 study utilized a stratified random sampling approach of individuals between the ages of 25 and 74 from specific regions of Finland (Supplementary Fig. 1). These areas included North Karelia in the east, Northern Savonia in the east, Oulu in the northwest, the province of Lapland in the north, Turku and Loimaa regions in the southwest, and the cities of Helsinki and Vantaa capital region in the south. In addition, we used the West-East split of the regions based on

the broad demographic and genetic characteristics of the Finnish population; the Western subset covers the regions of Turku/Loimaa and Helsinki/Vantaa, and the Eastern subset covers the rest of the regions (North Karelia, Northern Savonia, Oulu, Lapland). The sampling procedure was stratified by sex, region, and 10-year age group, resulting in 250 participants in each stratum. For Northern Karelia, Lapland, and the cities of Helsinki and Vantaa, the strata of 65–74-year-old men and women were also sampled, each with 250 participants. The initial population sample comprised 13,500 individuals (excluding 64 who had died or moved away between sample selection and the survey), with an overall participation rate of 65.5% (n = 8798). Of the participants, n = 7231 individuals successfully underwent stool shotgun sequencing. Of those, 129 participants withdrew their consent from the THL Biobank at the time of the study. We excluded four individuals due to failed sequencing (<100 reads). Subsequently, n = 7095 participants (mean age 49 years, 55 % women) remained for unsupervised analysis. The participants were not compensated.

No ethnicity data was collected from the participants. In 2002, the Finnish population was approximately 98% ethnic Finns, and most foreigners were from Russia, Sweden, and Estonia[34]. In FINRISK, 1.3% of the participants did not have Finnish or Swedish as their first language. Participant sex was identified using the Social Security number. Due to a lack of external cohorts with sufficient microbiome profiling and long-term health data, we used two internal subsamples to achieve a 70/30 train-test split (N = 5000 and N = 2095). We used cross-validation to examine the robustness of the results within the cohort.

**Population density.** The address-level coordinates of the participants were mapped with the *sf* R package[54] v. 1.0.9. to a 1 km² population grid from 2005, obtained through the *geofi* R package[55] v. 1.0.7. of the participants' home addresses ranged from 1 to 19,175 inhabitants/km² (mean 1753 inhabitants/km²). The most densely populated regions are in Southern and South-Western Finland (in the cities of Helsinki and Turku, respectively). We classified the population density into five levels: (<10) 0–9 inhabitants/km²; (<100) 10–99; (<1000) 100–999; (<10,000) 1000–9999; (<20,000) 10,000–20,000. The data points were randomly displaced within a 5 km × 5 km grid to obscure identifiable addresses in the figures. The figures do not show addresses with a population density of less than 10/km².

**Household income.** Data was collected based on a questionnaire and was used as the primary demographic descriptor variable alongside sex and age. We also used education level (educational years adjusted for birth year, with the levels low, medium, and high) in the models.

**Hospital exposure.** Data was collected based on a questionnaire. Participants reported how many days they were in a hospital in the past year. 770 participants reported spending at least 1 day in a hospital.

**Ethical approval.** The study protocol of FINRISK 2002 was approved by the Coordinating Ethical Committee of the Helsinki and Uusimaa Hospital District (Ref. 558/E3/20 1). All participants signed informed consent. The study was conducted according to the World Medical Association's Declaration of Helsinki on ethical principles.

**Baseline examination.** The FINRISK 2002 survey included a self-administered questionnaire, physical measurements, and blood and stool sample collection. The questionnaire and an invitation to the health examination were mailed to all subjects. Trained nurses conducted physical examinations and blood sampling in local health centers or other survey sites. The participants were advised to fast for ≥4 h and avoid heavy meals earlier during the day. The venous blood samples were centrifuged at the field survey sites, stored at −70 °C, and transferred daily to the Finnish Institute for Health and Welfare

laboratory. Data was collected for physiological measures, biomarkers, dietary, demographic, and lifestyle factors.

**Stool sample collection.** All willing participants were given a stool sampling kit at the baseline examination with detailed instructions. The participants mailed their samples overnight between Monday and Thursday under Finnish winter conditions to the Finnish Institute for Health and Welfare laboratory, where they were stored at −20 °C. The stool samples were transferred frozen in 2017 to the University of California, San Diego, for microbiome sequencing.

### Stool DNA extraction and library preparation
A miniaturized version of the Kapa HyperPlus Illumina-compatible library prep kit (Kapa Biosystems) was used for library generation. DNA extracts were normalized to 5 ng total input per sample in an Echo 550 acoustic liquid-handling robot (Labcyte Inc.). A Mosquito HV liquid-handling robot (TTP Labtech Inc.) was used for 1/10 scale enzymatic fragmentation, end-repair, and adapter-ligation reactions. Sequencing adapters were based on the iTru protocol[56], in which short universal adapter stubs are ligated first, and then sample-specific barcoded sequences are added in a subsequent PCR step. Amplified and barcoded libraries were then quantified by the Pico-Green assay and pooled in approximately equimolar ratios before being sequenced on an Illumina HiSeq 4000 instrument to an average read count of ~900,000 reads per sample. 107 negative controls were sequenced with the samples. The metagenomic data are available from the European Genome-Phenome Archive (accession number EGAD00001007035).

### Taxonomic and ARG profiling from sequencing data
We analyzed shotgun metagenomic sequences using a pipeline built with the Snakemake[57] bioinformatics workflow library. We trimmed the sequences for quality and adapter sequences using Atropos[58]. We removed host reads by genome mapping against the human genome assembly GRCh38 with Bowtie2[59].

We performed taxonomic profiling using MetaPhlAn3 v 3.1.0 and MetaPhlAn4 v 4.0.6[60,61] for R1 and R2 reads using the default settings. We mapped the R1 and R2 reads with Bowtie2[59] v 2.4.4[59] against the ResFinder database version 2.1.1[62] with the following options: "-D 20 -R 3 -N 1 -L 20 -i S,1,0 5" to identify ARGs. ResFinder[62] was chosen since it is an ARG database that exclusively contains acquired and clinically relevant ARGs, as our focus was on these types of ARGs, not on efflux pumps or other genes with less clear roles in clinical AMR. The default quality scores in Bowtie were used to ensure high-quality matches are included[59]. SAMtools v1.10[63] was used to filter and count reads, and if both reads mapped to the same gene, the read was counted as one match, and if the reads mapped to different genes, both were counted as hits to the respective gene. ARG counts were normalized by library size (number of reads per sample), ARG length, and the sum of all normalized ARGs per kilobase per million reads (RPKM). Negative controls showed minimal contamination, with a median of 1 ResFinder hit per sample, compared to a median of 319 hits in real samples. The mapping results from negative controls show that there is no systematic contamination in the samples (Supplementary Data 4).

### Phylogenetic tree visualization of bacterial taxa and inferred ARG hosts
We explored the phylogenetic distribution of ARG hosts in the ResFinder4 database in public sequence data, as our shallow shotgun sequencing did not allow for assigning ARGs to their host genomes using our data. This was done by searching for ARG matches in the nt database that covers nucleotide sequence entries. The blastn command was run using the ResFinder4 database as the query and the nucleotide collection "nt" database[37] as a reference, filtering for *e*-value < 10⁻⁶ with custom 'outfmt 6' including 'taxid' for the taxonomic

identifier. The blastn results were processed using TaxonKit[64] to add genus information based on the identifier to match the genera found in our cohort using genus names. Genera found in MetaPhlAn3 mapping filtered using the mergeFeaturesByPrevalence function in mia for at least 0.1% abundance in 1% of the samples were used to build a phylogenetic tree of the prevalent genera using ggtree v.3.8.2[65]. The most abundant ARGs in the cohort and their presence in the genera were visualized on the tree using the *gheatmap* function from ggtree.

### Register linkage for pre-existing diseases and medication use at baseline

In Finland, each permanent resident is assigned a unique personal identity number at birth or after immigration, which ensures reliable linkage to the electronic health registers. The data contain sensitive information from healthcare registers and are available through the THL biobank upon submission of a research plan and signing a data transfer agreement (https://thl.fi/en/web/thl-biobank/for-researchers/application-process).

The Finnish health registers cover nearly 100% of all major health events (Hospital Discharge Register, since 1969) and all prescription drug purchases (Drug Purchase Register, since 1995). The quality of the diagnoses in the Finnish national registers has been previously validated[5,6]. Antibiotic drug usage was based on prescription drug purchases (Drug Purchase Register) with the Anatomical Therapeutic Chemical (ATC) class J01 (antibiotics), which we used as a proxy for actual antibiotic use. We used the number of purchases and a binary categorical variable for any purchases during the follow-up in the analyses.

Baseline recent antibiotics use ($n = 1246$) was defined as any purchase with an ATC code of J01 (including subclasses) up to 6 months before sampling, and prevalent antibiotic use as any registered antibiotic purchases during the seven years before sampling in 2002, which corresponds to available records (see also Supplementary Data 1 for the antibiotic classes analyzed). The cumulative number of total antibiotic drug purchases during the past seven years before baseline varied from 0 to 85 (mean 3.3; Supplementary Fig. 1). Very recent baseline antibiotic use (<1 month prior to sampling) was treated separately from baseline use. Many antibiotics had very few individuals with recent purchases and thus were later filtered out due to having near-zero variance (see also "Statistical analysis" and "Participant data and variable preprocessing" sections). The participants were followed through Dec 31, 2019.

As a preliminary analysis of the data, we associated the use of the common antibiotic classes with the respective antibiotic resistance class ARG load (Supplementary Table 12). All these antibiotics correlated significantly with the respective resistance (log10 linear model).

Penicillin and other beta-lactam-antibiotics were purchased most often (5390 and 5529 unique purchases in the cohort during 7 years of recording before sampling). Tetracyclines were purchased 5179 times; macrolides, lincosamides, and streptogramins 4620 times. There were no purchases of aminoglycosides (Supplementary Table 1).

### Food questionnaire

Habitual diet was assessed using a food propensity questionnaire (FPQ), which contained 42 food items with choices ranging from 1 to 6 for consumption frequency. Answers denote the following descriptions: An answer 1 ("Less than once a month") 2 ("Once or twice a month") 3 ("Once a week") 4 ("Couple of times a week") 5 ("Almost every day"), and 6 ("Once a day or more often"). For raw vegetable and salad consumption, the answers 1 and 2 were combined, resulting in new levels 1 (Less than twice a month), 2 ("Once a week"), 3 ("Couple of times a week"), 4 ("Almost every day"), 5 ("Once a day or more often"). For poultry, levels 5 and 6 were combined, resulting in a new level 5 (Almost every day or more often), but the other levels were kept the same as in the original questionnaire. Additionally, the healthy food score[19], HFC was used as a proxy for the general healthiness of the diet.

### Regional analysis

For regional analysis, we used the six geographical regions defined above (North Karelia, Northern Savonia, Turku and Loimaa, Helsinki and Vantaa, Oulu, and Lapland) or the East-West split of the regions.

### Statistics & reproducibility

The analyses are based on previously collected data; therefore, we did not use statistical methods to predetermine sample size. No data that passed the minimum sequencing criteria were excluded from the analyses. The experiments were not randomized, as this was an observational study. The investigators were not blinded to allocation during experiments and outcome assessment.

**Participant data and variable preprocessing.** We excluded all variables with near-zero variance (*caret* R package[66] v. 6.0-94) or more than 500 missing values. The ARG load was log10-transformed for all the statistical analyses. The dichotomous variables and variables with less than ten levels were unscaled, and other variables, excluding ARG load, were scaled.

**Statistical analysis.** All statistical analyses were done in R[67] version 4.3.1. We corrected for multiple testing using FDR correction (Benjamini–Hochberg, R *stats* package). We report the adjusted P unless stated otherwise. We considered an FDR-corrected $P < 0.05$ significant. We report all *P*-values smaller than $1 \times 10^{-3}$ in the main text as $P < 1 \times 10^{-3}$. Exact *P*-values are given in Supplementary Data and Supplementary Information when possible. All figures were created with *ggplot2*[68] v. 3.4.4 unless otherwise indicated. For all analyses, including microbial taxa, the taxa abundances were centered log-ratio (CLR) transformed to account for compositionality unless otherwise indicated.

**General cohort statistics.** ARG load was measured using the total sum of all ARGs' reads per kilobase per million mapped reads (RPKM). The RPKM values varied considerably among the participants, with a range of 4.3–2607 (mean 272 RPKM). The total number of ARG reads mapping to the ARG database had a mean of 468 per sample. There was no association between library size and ARG load ($P = 0.47$, linear model). ARG load saturated with the used sequencing depth (see Supplementary Fig. 9 for sequencing depth and ARG metric variation along read counts).

**Alpha diversity.** We characterized the alpha diversity of the microbiome with the Shannon index using the complete species-level abundance data for the taxonomic profiles and using the complete ARG abundance data for the resistome profiles. The diversity of ARGs, as measured by the Shannon diversity index, ranged from 0.0 to 4.5, with a mean of 2.8. The number of unique ARGs detected ranged from 1 to 194, with a mean of 42. ARG Shannon diversity saturated with the used sequencing depth (Supplementary Fig. 9). ARG richness plateaued at around 3–4 million reads (Supplementary Fig. 9), suggesting that it did not saturate at the mean sequencing depth of 900,000 reads. ARG richness was not incorporated in the models for this reason.

**Verification of main findings, excluding low sequencing depth samples.** Supplementary Fig. 9 shows greater variability in the ARG metrics at lower sequencing depths. To address this, we replicated the GLM analysis for ARG load and diversity; the covariates selected by the boosted GLMs for the entire dataset (see Fig. 2) were also applied to a model that excluded samples with low sequencing depth (Supplementary Fig. 10). This ensured that these samples did not influence the results of the boosted GLMs. The covariates were ordered by their

estimates, consistent across both figures, Fig. 2 and Supplementary Fig. 9, and the estimates remained qualitatively similar. This reinforces that the samples with the lowest read depth do not skew the main findings of the boosted GLMs.

**Beta diversity.** We used the standard combination of (non-linear) principal coordinate analysis (PCoA) based on the Bray–Curtis dissimilarity index (estimated with the R packages scater[69] v1.29.4 and vegan[70] v2.6-4) to visualize the overall population variation of the microbiome and resistome composition. The beta-diversity analysis for taxonomic composition was based on species-level relative abundance data from MetaPhlAn3[60]. The beta-diversity analysis for resistome composition was based on the ARG profiles.

**Selection of covariates for modeling.** The covariates included in log-linear models were chosen based on 1148 available covariates. We removed covariates that defined events after sampling and had at least 500 missing values or near-zero variance (R package caret v6.0-94, nzv function with default settings), yielding 134 covariates (Supplementary Data 1) that included prior disease diagnoses for major non-communicable diseases and drug purchase events, as well as geographic region, sex, age, and food frequency. The diseases that passed the filtering criteria included high blood pressure, asthma, diabetes, skeletal fractures, ischemic heart disease, and major cardiovascular events. The majority of diseases did not pass the near-zero variance criterion described above. Two categories for drug purchases were used to investigate whether the association differed between recent (6 months) and prior use (7 years).

**Log-linear models.** We performed log-linear models pairwise with ARG load and ARG diversity, and all the explanatory variables that fit the selection criteria. Antibiotic use was controlled using the following parameters: number of events treated with all antibiotics and tetracyclines, prevalent MLSB and tetracycline use, and use of any antibiotic during the past month before baseline. The exponentiated estimate transformed to percentages of change for ARG load is reported in the main text for ease of interpretation. For example exponentiated estimate value 1.05 corresponds to a 5% increase in ARG load per unit of change in the covariate. The collinearity between key variables is quantified by pairwise Pearson correlations (Supplementary Table 9).

**Boosted GLMs.** Boosted generalized linear models (GLMs) with Gaussian distribution were fitted to associate ARG load (and, separately, ARG diversity) with covariates using the R packages mboost[71] v2.9-8, and caret[66] v6.0-94. We followed the same selection for boosted GLMs as for pairwise log-linear models. We further excluded the general diagnosis for mental diseases as it overlapped with drug purchases. Drug purchases were included as both prior (since 1995) and as baseline (past six months or past month) to investigate both short and long-term associations. Several variables, such as income class, sex, and food frequencies, exhibited collinearity (Pearson correlation, $P < 0.05$ for several comparisons; Supplementary Table 9). Despite this, each variable explains additional variation that is not captured by the other covariates.

In boosted GLMs, the generalized linear model is fitted using a boosting algorithm based on component-wise univariate (generalized) linear models. The variable selection is performed during fitting. Boosted GLMs adjust covariates iteratively. This makes them potentially more robust to the ordering of the covariates, and they can identify more optimal solutions, especially in high-dimensional settings. This also enhances interpretability compared to other standard approaches, such as logistic regression, that rely on stepwise updates[72]. The regression coefficients can be interpreted as regular GLM covariates. We reported the exponent of the coefficients

transformed to percentages of change for models in the main text for ease of interpretation.

The generalizability of the fitted models was assessed with cross-validation. Five thousand randomly selected participants from the cohort were used for model training, while the remaining 2095 individuals were used for testing to obtain a 70/30 train-test split. We excluded variables such as height and triglycerides, collinear with BMI, diet, and sex, which were the key variables of biological interest as part of standard data filtering before model training.

**Accounting for microbiome composition.** To better understand the role of microbiome composition in the associations between sex, geography, population density, and ARG load, we also fitted the boosted GLM with the relative abundances of the most prevalent bacterial families (those with an abundance above 0.01% in 1% of samples) as covariates, alongside the other covariates. The full model incorporating all covariates as well as bacterial family abundances identified with MetaPhlAn3 is presented in Supplementary Table 10 (see also Supplementary Table 11 for a supporting analysis with MetaPhlAn4). Nevertheless, the inclusion of so many covariates caused some diet and geographic covariates to lose statistical significance. Therefore, we conducted additional linear model analyses with fewer covariates alongside our model to assess the role of microbiome composition in specific associations. First, we ran a linear model including the relative abundance of bacterial families, poultry consumption, raw vegetable intake, and antibiotic use to examine whether the associations of ARG load with poultry and raw vegetable consumption still held when adjusting for microbiome composition. Similarly, we ran separate linear models including bacterial families, sex, East Finland residence, population density, and antibiotic use to assess whether the association of ARG load with sex, geographic region, and population density persisted independently of microbiome composition.

**Association between ARG load and covariates.** In order to detect potentially non-linear trends in ARG load concerning key covariates (raw vegetable consumption, poultry consumption, household income, population density, age) and to quantify uncertainties (Fig. 2c), we implemented a probabilistic model to predict mean ARG load. We modeled the relation between ARG load and each factor level in the given covariate based on the lognormal distribution using the default values in the R brm function from the brms package (version 2.21.0). The model can be summarized in the following pseudocode: brm(ARG load-factor(variable)-1, family = lognormal()). We ran this model separately for each sex. Posterior simulations were used to estimate the mean and credible intervals at the top 5% quantiles for the lognormal distribution at each factor level.

**Enrichment analysis.** We estimated enrichment of high-ARG individuals (top-10% quantile; >458 RPKM) as a function of enterosignature abundance (Fig. 3c) and relative abundance of dominant bacterial families (Supplementary Figs. 7b, d). In order to detect potentially non-linear trends in high-ARG enrichment, we estimated the enrichment separately in five distinct abundance bins (B0: not detected; B1–B4: 25% abundance quartiles among individuals with detected signal). For each abundance bin, we estimated the prevalence of high-ARG individuals using a probabilistic Bernoulli model with a logit link and an uninformative Gaussian prior N(0, 10) using the R brm function from the brms package (version 2.21.0). The model can be summarized in the following pseudocode: brm(High ARG-Bin - 1, family = bernoulli(link = "logit"), prior = prior(normal(0, 10))). As a validation step, we confirmed that the estimated prevalence from this model aligned with the observed prevalence in each abundance bin as expected. However, the probabilistic treatment allowed us to additionally estimate uncertainty and obtain more robust prevalence estimates for bins with a small

sample size; posterior simulations were used to estimate the mean and 95% credible intervals for the estimated prevalence at each bin. In order to evaluate enrichment of high-ARG individuals in each abundance bin, we compared the estimated prevalence with the expected prevalence (10%, i.e., the high-ARG individuals in the entire study population).

**Survival analysis.** We used the Cox proportional hazards model to predict all-cause mortality during the 17-year follow-up after baseline. The Cox regression model is a widely used method for analyzing time-to-event data. It estimates effect sizes while accounting for censored observations – data points where the event has not occurred by the end of the observation period. We inferred the model parameters with a probabilistic multivariate Cox model using the brms v 2.20.4[73] and tidybayes v3.0.6 R packages. We used the ggfortify[74,75] v0.4.16 and survminer[76] v0.4.9 packages to generate the Kaplan-Meier curve. We verified the probabilistic analyses with frequentist analyses based on the survival[77,78] v3.5-7 R package. Survival analysis was controlled for *Enterobacteriaceae* abundance, which we previously reported to associate with increasing mortality risk in this cohort[43], other mortality-associated covariates used in that publication (age, smoking, sex, diabetes, use of antineoplastic and immunomodulating agents, body mass index, self-reported antihypertensive medication), and raw vegetable consumption and income class were included as controls. Income and raw salad and vegetable consumption were adjusted for in the model since they have been negatively associated with mortality[29,44] but positively with ARG load. *Enterobacteriaceae* abundance and total ARG load were log10-transformed before the analysis. The median hazard ratio (HR) in Fig. 4a shows a relative change in mortality risk following a unit change in each covariate based on the probabilistic multivariate Cox regression model. The covariates that exhibited association with mortality are shown (>95% Bayesian credible intervals do not include zero). In order to compare effect sizes between continuous variables in the multivariate Cox models, we additionally scaled the continuous variables by their standard deviation. The Kaplar–Meier curves (Fig. 4b) compare survival between the individuals with high vs. conventional ARG load (the top-10% quantile vs. others; >458 RPKM), controlled for the same covariates as in the Cox regression model. The classical multivariate Cox model further confirms the association ($N = 7095$ participants, $P = 0.02$).

**Enterosignatures.** We adapted recently proposed enterosignatures to summarize the community composition in a few coherent subcommunities[41]. The enterosignature approach was proposed to complement the earlier attempts to stratify each individual into one of the few distinct community types driven by major groups of gut bacteria. In summary, we applied non-negative matrix factorization (NMF) on the genus-level relative abundances after combining the rare genera (1% prevalence above 0.1% relative abundance) into a single group ("Other"). We run NMF with 2–10 components with 10 runs using the default parameters in the function nmf from the R package NMF (v. 0.26). The Silhouette consensus measure in the function output indicated an optimal solution of 5 NMF components. Frioux et al. (2023) originally reported the same optimal number of components. We observed a notable correspondence of these signatures between FINRISK (Supplementary Fig. 5) and Frioux et al. (2023; Fig. 1). Three of the components had the same most abundant genus in both cases (*Bacteroides*, *Prevotella*, *Bifidobacterium*). In the *Firmicutes* component, the most abundant ten genera were *Firmicutes*, which accounted for 94% of this component in FINRISK. The *Escherichia* component was dominated by *Butyrivibrio* in FINRISK, with fewer Proteobacteria in the FINRISK data. Yet, it was the only signature observed in FINRISK associated with *Escherichia* (a scaled component loading 100% for this signature). These results were obtained with taxonomic profiles derived with MetaPhlAn3. We further verified the analysis using taxonomic profiles derived with MetaPhlAn4, which yielded 274 genera with the same prevalence filtering. Replicating the analysis with five enterosignatures yielded a significant ($P < 10^{-3}$; degrees of freedom = 7093) correspondence between MetaPhlAn3 and MetaPhlan4, as quantified by Pearson correlation of the ES abundance across all samples: ES-Bact ($r = 0.87$), ES-Firm ($r = 0.37$), ES-Prev ($r = 0.82$), ES-Bifi (0.84), ES-Esch (0.72). Notably, despite the differences, including independent datasets, metagenomic preprocessing pipelines, and implementation details, the five ES identified in FINRISK had a direct qualitative correspondence with the initially reported enterosignatures. The enterosignature abundances were robust to variations in library sizes (Kendall's tau; $P > 0.05$ for all ES; degrees of freedom = 7093).

**Kendall's rank correlation.** Associations between diversities, ARG load, and bacterial families or enterosignatures were calculated using Kendall's Rank Correlation (also known as Kendall's Tau), followed by FDR correction using the *mia*[79] package v. 1.9.19 function getExperimentCrossAssociation.

### Reporting summary
Further information on research design is available in the Nature Portfolio Reporting Summary linked to this article.

## Data availability
The metagenomic data are available from the European Genome-Phenome Archive (accession number EGAD00001007035, https://ega-archive.org/studies/EGAS00001005020). The phenotype data are available under restricted access as they contain sensitive information from healthcare registers and are available through the THL biobank upon submission of a research plan and signing a data transfer agreement (https://thl.fi/en/research-and-development/thl-biobank/for-researchers/application-process).

## Code availability
The source code for the analysis is available at https://doi.org/10.5281/zenodo.15574151.

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

## Acknowledgements

CSC IT Centre for Science computational resources were used for bioinformatic analysis of the samples. Shivang Bhanushali is acknowledged for participating in proofreading the manuscript. Research Council of Finland grant 348439: K.P. Research Council of Finland grant 338818: M.O.R. Research Council of Finland grant 340314: G.S.K. Research Council of Finland grant 321351: T.N. Research Council of Finland grant 354447: T.N. Research Council of Finland grant 330887: V.L. and L.L. Alhopuro foundation grant 20220114: K.P. Alhopuro foundation grant 20210172: G.S.K. Australian National Health and Medical Research Council (NHMRC) grant GNT2013468: C.G.V. and G.M. Finnish Cultural Foundation grant 210944: M.O.R. Sigrid Jusélius Foundation: T.N.

## Author contributions

Writing - original draft: K.P. Writing - final version: K.P., G.S.K., and L.L. Writing - review and editing: M.O.R., V.L., P.K., G.M., C.G.V., M.I., R.K., V.S., A.S.H., and T.N. Conceptualization: K.P., V.S., T.N., A.S.H., and L.L. Supervision: L.L. Data analysis: K.P., M.O.R., V.L., and L.L. Data acquisition: R.K., V.S., A.S.H., and T.N.

## Competing interests

R.K. is a scientific advisory board member and consultant for BiomeSense, Inc., has equity, and receives income. He is a scientific advisory board member and has equity in GenCirq. He is a consultant and scientific advisory board member for DayTwo and receives income. He has equity in and acts as a consultant for Cybele. He is a co-founder of Biota, Inc., and has equity. He is a co-founder of Micronoma and has equity and is a scientific advisory board member. The terms of these arrangements have been reviewed and approved by the University of California, San Diego, in accordance with its conflict of interest policies. Other authors declare no competing interests.
