## [Transparent Peer Review file · Nature Communications]

Variation and prognostic potential of the gut antibiotic resistome in the FINRISK 2002 cohort

Corresponding Author: Dr Katariina Parnanen

Version 0:

Reviewer comments:

Reviewer #1

(Remarks to the Author)
General Comments:

The authors evaluate correlations between antimicrobial resistance gene (ARG) abundance in the gut microbiome and demographic factors in over 7000 Finnish people sampled in the year 2002. They find that in addition to the known risk of antibiotic use, ARG load/relative abundance was also increased among adults who consumed fresh vegetables and poultry, were female, were from higher income groups or who lived in urban areas. This study addresses an important issue of understanding risk factors which drive the increasing antimicrobial resistance rate seen globally. Notably, they also find that ARG abundance predicts mortality as well as blood pressure, and also increases sepsis risk. The main flaw of this study is the homogeneity of the Finnish population and the generalizability to other regions/countries. In addition, while the 17-year follow-up allows for prediction of long-term outcomes, significant changes have occurred in the last two decades with regards to baseline antimicrobial resistance, increased immigration and diversity within Finland, and likely differences in agricultural practices regarding antibiotics. Performing analyses at the individual ARG class level (versus simply total ARG load) would add a more informative and comprehensive dimension to this study. All said, this is a well-designed population-level cohort study with significant and novel findings that can inform our public health efforts to combat antimicrobial resistance.

Specific Comments:

Abstract:

1. If word count allows, the authors should include the more important quantitative results within the abstract to better capture the degree and the strength of the linkage between ARG load and risk factors such as consumption of fresh vegetables and poultry, female sex, and urban high-income demographics.

Introduction:

1. Race/ethnicity data should be included when describing the population cohort.
2. Line 53: Typo "...one of they driver..."
3. Line 68: Typo for "Supplementary Fig" should be a hyperlinked "Supplementary Fig. 1".

Results:

1. It would be informative if the authors "Drug use" section:
 - a. How did the authors choose a seven-year exposure history for antibiotics?
 - b. Did the authors evaluate time of last antibiotic exposure to ARG load?
 - c. Did the duration of antibiotic exposure or number of antibiotic purposes or number of different types of antibiotic classes affect the ARG load?
2. "Diet" section:
 - a. Did the authors evaluate how different amounts of raw vegetables and salad consumed affected the ARG load?
 - b. Line 126: Are there any data on the types of ARGs found in poultry vs beef or pork at this time? It would be interesting to see if the ARGs found in this study correlated with the ARGs circulating in poultry.

- c. Line 142: It is not clear that the findings do support direct selection over indirect selection as different food products can drive different microbiomes, and indirectly affect the ARG loads. A more conclusive argument would be if the authors tested the vegetables and poultry and found the same ARGs in the food products as in the human microbiome, or if the authors demonstrate that the gut microbiome (or enterosignatures) didn't differ between people with different diets.
3. The authors should briefly mention how enterosignatures were identified while introducing them in the results.
 4. How did the authors associate ARGs with specific taxa as shown in 3d?
 5. "Gender and income" section: Line 154: It is true these are gender-specific differences, but some (such as prevalence of UTIs) could be linked mechanistically to increased ARG loads primarily through more antibiotic use.
 6. "Geographic variations" section: Are there differences in characteristics such as proportion of immigrants, travel exposure, etc. between urban and rural regions that may also contribute to the difference in ARG loads seen between the regions?
 7. "The role of the gut microbiome in shaping the resistome" section: Did the authors review the ARGs detected to see if ARGs of primary public health concern (such as CTX-M, or carbapenemases such as NDM, KPC, OXA-48, etc.) were carried by specific bacterial genera?
 8. "Long term mortality" section: Did the authors use data from the 2002 survey (at the time of the stool collection for metagenomic sequencing) or from a later survey (closer to the time of death or at the end of the 17-year follow up) as covariates to predict long-term mortality and sepsis risk?
 9. This is out of the scope of this adult-focused study, but do the authors plan to carry out a similar study in Finnish children?
 10. Can the authors perform correlation analyses between specific bacterial genera and total ARG load as well as individual ARG load?
 11. Did hospital exposure correlate with ARG load?

Methods:

1. Can the authors include results from their primary analyses at the individual ARG class level (e.g., beta lactam, etc.?). For instance, does exposure to a specific antibiotic class (e.g., tetracycline) affect specific ARG classes (tetracycline resistance genes?)
2. Did the authors include negative or positive controls when carrying out their metagenomics?
3. The authors should provide additional details about the exact antibiotics included in the ATC antibiotic classification classes (e.g., JO1) as they are not widely used in the clinical infectious disease or microbiome literature
4. Can the authors elaborate in the manuscript on why they chose a boosted GLM model to identify factors including ARG abundance as opposed to a more widely used approach such as logistic regression?
5. Did the authors implement any QC thresholds with respect to ARG detection (e.g., minimum percent coverage of ARG, minimum number of reads, etc.)
6. How did the authors adjust for background contaminating environmental taxa or ARGs?

7. Discussion:

8. Typo line 262: "comparable predator"
9. The authors should include a brief discussion of limitations of the study. For instance, homogeneity of the Finnish population and possible lack of generalizability to other regions/countries, cross sectional analysis conducted over 20 years age.

Typo Line 476 Supplementaty

(Remarks on code availability)

Reviewer #2

(Remarks to the Author)

The manuscript is pretty large population-based analysis of population factors driving ARG load in gut microbiomes. While being a large study in population, this is not the only study based on microbiomes of FINRISK cohorts, and it might be improved in some aspects.

1. The core of the manuscript is the estimation of determinants of microbiome ARG on a population level. And one of observations immediately visible from Figure 2 are not the effects of phenotypes, but a terrific difference in ARG load between men and women. In population-based studies, sex is usually taken into account as a "default" covariate. Although, I think, this dataset might be great for having a deeper look into gender aspect of ARG load. What kinds of sex-determined host properties make the ARG load so different between men and women? In the manuscript, there's a lot of examples of association analyses of X to Y conditioned to Z. I guess the sex ARG story might be addressed in a similar way, without necessity for a specific research line. Just some extra reflection and conditional analyses.
2. There's absolutely no excuse for using metaphlan3. It's just outdated. It's worth switching to metaphlan4, especially given the fact that ESEs might change substantially.
3. As a validation, it would be good to check the concordance between ResFinder and CARD. Not necessarily repeat the whole analysis, but to show how those two databases agree
4. One of the typical problems of ARG calling in metagenomic data is a strict dominance of efflux pump genes, which might act as ARGs but might also have alternative metabolic functions. When working with CARD database, it might be

recommended to remove those genes from analysis, or at least treat them separately.

5. In principle, the idea of classifying ARGs based on their type (not only the host or targeted antibiotic) might also be a good idea to explore in regards to phenotype associations.

Overall, it's a great manuscript and my kudos to authors.

(Remarks on code availability)

Reviewer #3

(Remarks to the Author)

In this manuscript, the authors aimed to investigate socio-demographic and gut microbiome factors that drive resistome variations. To this end, the authors characterized the gut resistome from fecal samples collected in 2002 of 7,095 adults from six contrasted Finnish regions. Metadata of this cohort includes address-level geographic location, diet, household income level, prescription drug purchases, diseases, and causes of death until 2019. Using supervised machine learning models, the author state that antibiotic usage and consumption of raw vegetables and poultry were positively correlated with the total ARG load. The authors also claim that resistance abundance was generally higher in females and urban high-income demographics. Moreover, using the Cox proportional hazards model, the authors predicted all-cause mortality during the 17-year follow-up period. They report on associations between ARG load and all-cause mortality, mortality by respiratory causes, and sepsis.

The human gut resistome is known to be closely related to human health. Researchers from different groups have reported various factors contributing to resistome variations, including age, sex, diet, antibiotic administration, socioeconomic status, and location. While the scientific question addressed in this study is not entirely novel, it stands out due to its use of a large Finnish cohort with comprehensive metadata. A key finding of this study is the correlation between resistance burden and mortality and sepsis risk. However, there are concerns regarding the collinearity of model features (as the authors mentioned in line 738-740) and inconsistencies in feature control across different models, which needs further clarification in the main text. Additionally, the discussion requires a more thorough exploration of the results and a more detailed evaluation of the advantages and limitations of the computational models.

Major Critiques

1. Sequencing depth for ARG profiling:

a. Capturing resistome composition and diversity requires sufficient sequencing depth, which should be empirically determined by rarefaction/ROC analysis. In the methods (Line 612), the average read count is described as ~900,000 reads per sample, which seems quite a bit lower than other published resistome studies. Notably, lines 558-559 mentions that “four individuals had zero reads mapping to the ARG database,” which raises the question of whether this is due to insufficient sequencing depth. To address this, the authors should provide the read count for each sample and include rarefaction analysis to demonstrate that the sequencing depth is adequate.

b. Line 681-682: “the ARG load for each participant ranges from 4.3 to 2607”. Does this variation correlate with read number? Again, having some form of rarefaction analysis would be useful for understanding whether these resistome comparisons are statistically valid.

2. ARG host prediction: It is noteworthy that no ARGs were assigned to Staphylococcaceae in either DataS1 or Fig 2c, a family of bacteria that is typically known to carry ARGs. I understand that the authors chose to use computational tools to predict the bacterial host due to the shallow shotgun sequencing. While the pipeline is well described in the methods section, the authors should discuss whether the absence of certain bacterial families carries biological significance. Additionally, they should address the limitations and potential biases of the tools used and consider how these factors might influence the conclusions. A validation analysis comparing computationally inferred ARG-host associations with known literature or database references would strengthen confidence in the findings.

3. GLM models: There were some inconsistencies in feature control during model prediction, which could be a concern due to collinearity between the features. For example, “the difference between genders remained significant even after controlling for differences in antibiotic use, diet, and the relative abundance of bacterial families” (lines 153 and 160) did not adjust for age, population density income, or demographic factors, and “Eastern Finns had a generally lower ARG load, a result that was robust to controlling for diet, health, population density, and demographic factors” (line 168-170) did not control for sex or antibiotic use. Please correct me if I misunderstood. If not, please clarify if it's necessary to exclude these features from the modeling. It is important to clarify why certain covariates were omitted and whether their inclusion would alter the reported associations. A sensitivity analysis including all major predictors in a unified model would improve robustness.

4. The authors use ‘gender’ as a variable, when I think they mean ‘sex’, as a biological variable. I interpret ‘gender’ to mean a spectrum of socially constructed roles, identities, and behaviors, whereas as I interpret ‘sex’ to mean biological characteristics defining a person to be male, female, or other. What do the authors intend? Please clarify.

5. Survival analysis:

a. As previously mentioned, one of the key highlights of this study is the correlation between survival events and the gut resistome. The authors used the Cox Proportional Hazards model to analyze survival times. It would be helpful if the authors could provide more details on the advantages of this tool. Also, since the input of ARG data was collected at a single time point, will it lead to any bias in the model prediction considering the resistome would change over time?

b. Please provide more discussion on the apparently conflicting results of higher ARG load in women and high-income populations but higher mortality rates in men and urban populations if ARGs are a risk factor for mortality (line 267-268).

c. This might be out of scope, but it would be interesting to detect the correlation between different ARG classes and mortality risk. (minor)

Minor Critiques

1. The statement in the abstract is vague due to the excessive use of adjectives, such as "healthier" and "higher sepsis risk." It would be more effective to provide specific data or numerical values to define these terms.
2. Line 30 "humans ,", delete the extra space before comma
3. Line 53, change "one of they driver rates" to "one of the driver rates"
4. Line 87, please state the details of "sometimes non-significant trends" and try to avoid using words like "sometimes" when talking about significance.
5. In line 96-102, the authors need to describe how they calculated 55% higher ARG load and other values in this paragraph. It's hard to figure it out from the Data S1 for the readers.
 - a. What are PREVAL_RX_J01A and PREVAL_RX_J01A_NEVT in Data S1? Please provide clear descriptions of the data provided.
 - b. Same for the 4% average increase per consumption level in line 117
6. Line 104, "respiratory medication (ATC class R)" delete one space between n and (
7. Line 136-139, "This might be explained by the lack of raw vegetables and poultry in the typical diets containing these foods. Nevertheless, such unhealthy diets are also associated with higher antibiotic consumption and disease prevalence in the population, which may confound the observed response of ARG-carrying taxa." Please provide data or figures to support this statement.
8. Line 153 and 160, are the two P values in the parentheses before and after controlling for bacterial abundances? Please clarify it in the text.
9. Line 194: I did not see Fig 2b correlate to the $R^2=0.001$ data. Please provide the correct figure.
10. Fig 3c: Please specify the methods/steps of ARG enrichment calculation. Add error bars to indicate confidence correlated to the sample size variations.
11. Line 213: What does "non-monotonic relations between enterosignature abundance and ARG load" indicate?
12. Line 262, fix "comparable predator"
13. Line 268-274: This paragraph is difficult to follow. Providing additional details would help improve understanding.
 - a. What are "these protective effects?"
 - b. How do we understand "early warning signals of the predicted shift?"
14. The authors utilized "individual level" a couple of times, such as in line 251 "socio-demographic determinants with individual-level resolution" and line 254 "underlying individual resistance levels." Although the cohort collected comprehensive metadata from each individual participant, the model prediction of ARG load or mortality risk was still over groups of instances with certain probability ranges. Please clarify the definition of this concept to avoid confusion.
15. Please correct the text font from line 584 to 602

(Remarks on code availability)

Version 1:

Reviewer comments:

Reviewer #1

(Remarks to the Author)

Overall, this is a very well done study that represents an important contribution to the literature, and I commend the authors on their work. I have a few final comments:

1. The authors state, "As a preliminary analysis of the data, we associated the use of the common antibiotic classes to the respective antibiotic resistance class ARG load (Supplementary Table 12). All these antibiotics correlated significantly with the respective resistance (log10 linear model). (lines 767-769). In Table S12, have the FDR > and < symbols been inadvertently switched?"
2. I recommend that the authors include a supplementary data file or table summarizing the ARG and taxonomic alignments from the 107 negative control samples for full transparency. It is standard practice to use some type of background-correction approach to statistically subtract or remove contaminants derived from the laboratory environment or reagents; since this was not done a table of alignments from the negative controls would be a reasonable alternative that would increase confidence in the presented results.
3. I suggest that the authors assess whether hospital exposure correlates with ARG load, as recommended before, as this is an important analysis that can be carried out with the available data.
4. I have had the chance to review the authors' response to reviewer 3, point 1:

The sequencing depth is indeed low for a metagenomics study. I can appreciate the authors' argument from the new Supp. Fig. 9 that ARG RPKM and diversity appear to saturate at 900k reads. However, they aren't using 900k reads as a threshold

for inclusion, it's their average. So for a large proportion of their data, it's likely not meeting this threshold for saturation.

There also appears to be very high variation in those metrics at lower sample read numbers as one might expect. So, for the samples that have with fewer reads, those data appear to be less reliable or interpretable.

To increase confidence in their findings, the authors could: 1) randomly subsample the dataset and determine the threshold below which FPKM changes significantly or 2) demonstrate that their significant findings are robust to the removal of low read count samples.

Sequencing read count is not identifiable information by any criteria I'm aware of, so to increase transparency about individual sample read counts, the authors could simply provide a histogram with samples on the Y axis and read count on the X axis, or alternatively swap patient ID with numbers 1-NNN and provide the information as source data for Supp. Fig. 9.

Overall, I do think this is an important contribution to the literature. These extra steps should be very feasible with their existing data and could provide additional confidence in the validity of the results.

(Remarks on code availability)

Reviewer #2

(Remarks to the Author)

I am satisfied with the responses given to my comments, and also to the comments raised by other reviewers

(Remarks on code availability)

Version 2:

Reviewer comments:

Reviewer #1

(Remarks to the Author)

The authors have thoroughly addressed my comments and concerns, and I commend them on an interesting and important study.

(Remarks on code availability)

made.

Rebuttal letter

REVIEWER COMMENTS

Reviewer #1 (Remarks to the Author):

General Comments:

The authors evaluate correlations between antimicrobial resistance gene (ARG) abundance in the gut microbiome and demographic factors in over 7000 Finnish people sampled in the year 2002. They find that in addition to the known risk of antibiotic use, ARG load/relative abundance was also increased among adults who consumed fresh vegetables and poultry, were female, were from higher income groups or who lived in urban areas. This study addresses an important issue of understanding risk factors which drive the increasing antimicrobial resistance rate seen globally. Notably, they also find that ARG abundance predicts mortality as well as blood pressure, and also increases sepsis risk. The main flaw of this study is the homogeneity of the Finnish population and the generalizability to other regions/countries. In addition, while the 17-year follow-up allows for prediction of long-term outcomes, significant changes have occurred in the last two decades with regards to baseline antimicrobial resistance, increased immigration and diversity within Finland, and likely differences in agricultural practices regarding antibiotics. Performing analyses at the individual ARG class level (versus simply total ARG load) would add a more informative and comprehensive dimension to this study. All said, this is a well-designed population-level cohort study with significant and novel findings that can inform our public health efforts to combat antimicrobial resistance.

However, there are concerns regarding the collinearity of model features (as the authors mentioned in line 738-740) and inconsistencies in feature control across different models, which needs further clarification in the main text. Additionally, the discussion requires a more thorough exploration of the results and a more detailed evaluation of the advantages and limitations of the computational models.

Thank you for your thorough review of our manuscript. We have carefully considered your comments and have now incorporated additional analyses and clarifications as requested.

We acknowledge that Finland has experienced demographic and agricultural shifts in recent decades. Nevertheless, the FINRISK study also presents advantages in this regard and is a well-characterized and highly structured cohort. Indeed, it provides a unique opportunity to examine the relationships between the resistome, diet, microbiome, population density and demographics with minimal interference from other population-level heterogeneity, such as genetic and cultural background. This makes our findings on patterns of resistome transmission and selection particularly robust within this context. Such a study is a valuable starting point that can later be contrasted with more heterogeneous populations to assess generalizability and broader global patterns. We have now expanded discussion on this in the Discussion section.

Regarding the concern about collinearity in model features (lines 738–740), we would like to clarify that excluding certain covariates does not affect the qualitative conclusions of our study. Prior to analysis, we systematically removed covariates that were highly correlated with other variables of interest to avoid multicollinearity and redundancy in the models. This is a standard prefiltering step in data analysis to ensure model interpretability and stability. We have added further clarification of this in the *Methods* section (lines 865-867).

For example, we excluded height and weight because they are strongly correlated with BMI, which is a more commonly used covariate in population studies. Similarly, we excluded triglycerides due to their correlation with diet and certain other metabolic markers. Our approach ensures that only the most biologically relevant variables are retained for model training. To address this concern, we have revised the manuscript to explicitly

state:

"We excluded variables such as height and triglycerides collinear with BMI, diet, and sex, which were the key variables of biological interest as a part of standard data filtering before model training."

Below, we have provided a detailed response to the possible inconsistencies in feature controls.

Additionally, we have expanded the discussion to provide a more thorough evaluation of our results, including the advantages and limitations of our methods. We appreciate the reviewer's feedback, which helped us improve the clarity and rigor of our manuscript.

Specific Comments:

Abstract:

1. If word count allows, the authors should include the more important quantitative results within the abstract to better capture the degree and the strength of the linkage between ARG load and risk factors such as consumption of fresh vegetables and poultry, female sex, and urban high-income demographics.

We have included more quantitative results in the abstract. We have opted to include the quantitative results regarding mortality, which was requested by Reviewer 3:

"Interestingly, resistance was not linked to the consumption of high-fat and high-sugar foods, but was consistently higher in females and urban high-income individuals, who currently have generally lower mortality rates. Nevertheless, during the 17-year follow-up, high resistance was associated with a 1.07-fold increase in mortality risk, comparable to elevated blood pressure, and with a heightened risk of sepsis."

Introduction:

1. Race/ethnicity data should be included when describing the population cohort.

Due to historical reasons, there is no central collection of ethnicity data in Finland, and at the sampling time, no specific race or ethnicity data was collected. The sampling represents a random sample of the Finnish adult population in 2002. Therefore, most participants are native Finns (approximately 98%), 1% from Russia, Sweden, and Estonia, and 1% from other countries. We have now described this more clearly in the *Methods* section *Study participant details*.

2. Line 53: Typo "...one of they driver..."

Thank you, we have corrected this.

3. Line 68: Typo for "Supplementary Fig" should be a hyperlinked "Supplementary Fig. 1".

Thank you - corrected.

Results:

1. It would be informative if the authors "Drug use" section:
a. How did the authors choose a seven-year exposure history for antibiotics?

This has now been detailed in the *Results* section *Drug use* and *Methods* section *Register linkage for pre-existing diseases and medication use at baseline*. Briefly, the seven years is defined by the establishment of the register, which collects drug purchase data in Finland (1995)

b. Did the authors evaluate time of last antibiotic exposure to ARG load? Did the duration of antibiotic exposure or number of antibiotic purposes or number of different types of antibiotic classes affect the ARG load?

We do not have other records of antibiotics besides purchase reimbursements. We have not estimated the effect of antibiotic use duration as this data is not available from the Drug reimbursement register.

The association of the number of antibiotic purchases and ARG load is analysed in linear models, (Data S1), and the new Supplementary Table 12 for specific antibiotic classes. We have clarified this on lines 757-758.

“We used the number of purchases and a binary categorical variable for any purchases during the follow-up in the analyses.”

We have categorized the time to last purchase the antibiotics to very recent (< 1 month before sampling), recent (< 6 months before sampling) and all prevalent use (Data S1). Some antibiotic classes had too few individuals with purchases in the recent and very recent categories, and these were filtered out from analysis due to having near zero variance in the data.

We have clarified the questions regarding the antibiotic purchases further in the Methods section Register linkage for pre-existing diseases and medication use at baseline.

In the case of macrolides, recent use < 6 months before sampling had a larger estimate than use during the 7 years of follow-up (Data S1). For all antibiotics, the recent and all use during follow up had similar estimates with overlapping 95% confidence intervals. We have now added these results to the main text (line 107 onwards) from the Data S1.

We did not analyse the effect of exposure to multiple classes of antibiotics on the ARG load. Future work on this topic in the FINRISK cohort would be worth pursuing.

We have not included further discussion on the antibiotic use comments in the discussion as we have covered in the Methods and Results section due to word limitations, further discussion could be added if seen fit in the Discussion section as well.

2. “Diet” section:

a. Did the authors evaluate how different amounts of raw vegetables and salad consumed affected the ARG load?

We do not have records of the amounts of food groups consumed, only the number of times they were consumed within a given period using a food frequency questionnaire (See reference Koponen, K. K. *et al.* Associations of healthy food choices with gut microbiota profiles. *The American Journal of Clinical Nutrition* **114**, 605–616 (2021).

b. Line 126: Are there any data on the types of ARGs found in poultry vs beef or pork at this time? It would be interesting to see if the ARGs found in this study correlated with the ARGs circulating in poultry.

Thank you - the FINRES-vet report in the references has detailed information of resistance of isolated bacteria from pigs, cattle and poultry. Overall resistance was more common in poultry isolates than in cattle or pigs. We have now included more information on food production animal resistance in the manuscript.

“In particular, tetracycline, macrolide, beta-lactam and amphenicol, which were all common antibiotic resistance classes in the FINRISK cohort (Supplementary Fig. 4), were common in poultry in 2002 but much less so in cattle according to surveillance reports²⁵.” lines 140-143

c. Line 142: It is not clear that the findings do support direct selection over indirect selection as different food products can drive different microbiomes, and indirectly affect the ARG loads. A more conclusive argument would be if the authors tested the vegetables and poultry and found the same ARGs in the food products as in the human microbiome, or if the authors demonstrate that the gut microbiome (or enterosignatures) didn't differ between people with different diets.

Unfortunately, we do not have access to food samples from 2002 or gene-level information on the resistance in poultry at the time. However, the antibiotic classes observed in poultry in 2002 according to FINRES-vet (reference 25) are also very common in our cohort samples.

We have added more analysis on bacterial family and ES and diet in the Supplementary Data S2 and included findings in the *Results* lines 147 onwards, 163-164, and 264-267. Our results suggest that some fiber-rich foods are associated with ES-Prevotella, which may mediate some of the findings regarding some fiber-rich foods associating negatively with ARG load.

3. The authors should briefly mention how enterosignatures were identified while introducing them in the results.

Thank you, we have modified the text to read:

“To further characterize co-variation between broad patterns of microbiome composition and resistance, we identified five sub-communities, or enterosignatures (ES) defined using non-negative matrix factorization (NMF)⁴¹.

4. How did the authors associate ARGs with specific taxa as shown in 3d?

Thank you for pointing out that this was unclear. We have detailed this more clearly in *Methods* line 737 onwards, and have now modified the text in the *Results* section (lines 230-233). Note that we have also updated the subfigure labeling (3d -> 3e):

“Furthermore, after identifying the range of possible bacterial hosts for different ARGs through a BLAST search against the nt database^{37,38}, we found phylogenetic clustering among the hosts of several prevalent ARGs (Fig. 3e, See Methods for more details on ARG host identification”

5. “Gender and income” section: Line 154: It is true these are gender-specific differences, but some (such as prevalence of UTIs) could be linked mechanistically to increased ARG loads primarily through more antibiotic use.

Thank you; we have included a mention that these may link to higher antibiotic use (lines 178-179).

6. “Geographic variations” section: Are there differences in characteristics such as proportion of immigrants, travel exposure, etc. between urban and rural regions that may also contribute to the difference in ARG loads seen between the regions?

We agree that this is likely also contributing to the variation between urban and rural areas. We have added a sentence to this section to elaborate on this topic. *“The densely populated urban areas of Turku and Helsinki also receive more international travelers and immigration, which contributes to the transmission of ARGs from abroad.*

.”

7. “The role of the gut microbiome in shaping the resistome” section: Did the authors review the ARGs detected to see if ARGs of primary public health concern (such as CTX-M, or carbapenemases such as NDM, KPC, OXA-48, etc.) were carried by specific bacterial genera?

We identified the typical hosts of common ARGs using nucleotide databases. Critical ARGs were rarely detected, as antibiotic resistance in Finland in 2002 was primarily limited to commonly used antibiotics. We could not predict their hosts because we could not assemble the data. We looked at the prevalence of the mentioned genes and they were either not detected or detected in only one sample, prohibiting correlation-based prediction of hosts.

We added discussion on the limitations of the study regarding the detection of rare ARGs (lines 322-325).

8. “Long term mortality” section: Did the authors use data from the 2002 survey (at the time of the stool collection for metagenomic sequencing) or from a later survey (closer to the time of death or at the end of the 17-year follow up) as covariates to predict long-term mortality and sepsis risk?

The ARG data is from 2002, and the covariate data comes from the 2002 survey or registers. For example, blood pressure was recorded in 2002, and for diabetes, we included prevalent cases, not those after 2002.

9. This is out of the scope of this adult-focused study, but do the authors plan to carry out a similar study in Finnish children?

Yes, we are planning to expand this to children in the future.

10. Can the authors perform correlation analyses between specific bacterial genera and total ARG load as well as individual ARG load?

The *Results* section (lines 261-264) now includes total ARG load and its association with bacterial families (Data S1, Family associations with ARG load). To provide a more robust analysis, we used families instead of genera.

11. Did hospital exposure correlate with ARG load?

This would be interesting to study in the future. However, given the time allotted for the review, we cannot incorporate this now. However, FINRISK only includes participants not in hospitals or other care facilities (such as nursing homes). See also reference Borodulin, K. et al. Cohort Profile: The National FINRISK Study. *International Journal of Epidemiology* 47, 696–696i (2018). We used data on prevalent diseases, which might correlate with hospital exposure. For example, asthma (Data S1) was correlated with ARG load even after controlling for antibiotic purchases.

Methods:

1. Can the authors include results from their primary analyses at the individual ARG class level (e.g., beta lactam, etc.?). For instance, does exposure to a specific antibiotic class (e.g., tetracycline) affect specific ARG classes (tetracycline resistance genes?)

We have added this to the *Methods* section and provided a new Supplementary Table 12. We analysed this for the most common antibiotics reimbursed and they all were significantly associated with higher ARG load for the respective ARG class (lines 767-769).

2. Did the authors include negative or positive controls when carrying out their metagenomics?

The sequencing contained 107 negative controls. Only minimal ARGs contamination occurring from sporadic hits to the ResFinder database was observed in the negative controls. This information has now been added to the *Methods*

"107 negative controls were sequenced with the samples." line 717

"Negative controls showed minimal contamination, with a median of 1 ResFinder hit per sample, compared to a median of 319 hits in real samples." lines 734-735

3. The authors should provide additional details about the exact antibiotics included in the ATC antibiotic classification classes (e.g., J01) as they are not widely used in the clinical infectious disease or microbiome literature

We have included clarification in the *Methods* section J01 = antibiotics (line 772) and Data S1 which has the regression results and now includes the common name for the antibiotic classes, for example J01F = macrolides.

4. Can the authors elaborate in the manuscript on why they chose a boosted GLM model to identify factors including ARG abundance as opposed to a more widely used approach such as logistic regression?

We have now added justification in the *Methods* section, including reference 75 (line 873 onwards).

"Boosted GLMs adjust covariates iteratively. This makes them potentially more robust to the ordering of the covariates, and they can identify more optimal solutions, especially in high-dimensional settings. This also enhances interpretability compared to other standard approaches, such as logistic regression, that rely on stepwise updates⁷⁵."

5. Did the authors implement any QC thresholds with respect to ARG detection (e.g., minimum percent coverage of ARG, minimum number of reads, etc.)

We used Bowtie2, which has an alignment score system that does not provide a direct option for setting a minimum percentage coverage or identity threshold. We used the default alignment score, which includes only high-quality matches with a high identity percentage by default—consistent with recommendations for ARG studies (Bengtsson-Palme et al., 2017, *Journal of Antimicrobial Chemotherapy*, <https://academic.oup.com/jac/article/72/10/2690/3904526>).

This has been now clarified in the *Methods* section (lines 740-745).

6. How did the authors adjust for background contaminating environmental taxa or ARGs?

The contaminating ARGs, which are environmentally transmitted ARGs, would be of interest to our research question. However, our participant data did not include covariates that would directly allow for the identification of such taxa or ARGs. This is a latent variable in our study that we did not model.

7. Discussion:

8. Typo line 262: "comparable predator"

Thank you, this has now been corrected.

9. The authors should include a brief discussion of limitations of the study. For instance, homogeneity of the Finnish population and possible lack of generalizability to other regions/countries, cross sectional analysis conducted over 20 years age.

Thank you, we have discussion detailing the limitations of the study (lines 319-346)

Typo Line 476 Supplementaty

Thank you, this has now been corrected

Reviewer #2 (Remarks to the Author):

The manuscript is pretty large population-based analysis of population factors driving ARG load in gut microbiomes. While being a large study in population, this is not the only study based on microbiomes of FINRISK cohorts, and it might be improved in some aspects.

1. The core of the manuscript is the estimation of determinants of microbiome ARG on a population level. And one of observations immediately visible from Figure 2 are not the effects of phenotypes, but a terrific difference in ARG load between men and women. In population-based studies, sex is usually taken into account as a "default" covariate. Although, I think, this dataset might be great for having a deeper look into gender aspect of ARG load. What kinds of sex-determined host properties make the ARG load so different between men and women? In the manuscript, there's a lot of examples of association analyses of X to Y conditioned to Z. I guess the sex ARG story might be addressed in a similar way, without necessity for a specific research line. Just some extra reflection and conditional analyses.

Thank you for the thorough review of the manuscript. We agree that the gender differences are interesting and have another manuscript in progress utilizing a larger dataset to explore these differences. We have now added additional linear model results with just women to Data S1. The associations followed similar trends as in the whole data, given that significance was lower for the lower sample size.

"The covariates most associated with ARG load in women were similar to those in the whole cohort (Data S1), with tetracycline use having the strongest effect, followed by respiratory drugs. " lines 179-181

2. There's absolutely no excuse for using metaphlan3. It's just outdated. It's worth switching to metaphlan4, especially given the fact that ESes might change substantially.

We agree that taking advantage of the most up-to-date versions of bioinformatic programs is generally best practice. MetaPhlAn4 is relatively new (published in February, 2023), and the taxonomic data we used in this study was generated before MetaPhlAn4 was available. Given the significant changes in the database and nomenclature in MetaPhlAn4, entirely redoing the taxonomic analyses with the new version would require major reanalysis and rewriting. This is not feasible given the time frame that is available for our response. Moreover, the presented taxonomic analyses are primarily intended to complement antibiotic resistance gene analyses.

Therefore, instead of changing the preprocessing method, we added complementary analyses to verify qualitative key results with MetaPhlAn4. The new analyses show that the main conclusions remain unaltered with this change. In particular, we analysed associations between ARG load, bacterial taxa, enterosignatures, and covariates using MetaPhlAn4:

- 1) community similarity (PCoA) and associations between ARG load and the relative abundance of dominant bacterial families are now shown for both data versions in the updated Figure S7; in addition to the original results with MetaPhlAn3 (Fig. S7a-b), we have added reanalysis with MetaPhlAn4 (Fig S7c-d). The broad patterns of the population distribution (e.g. high-Prevotella community type) can be

discerned in PCoA ordination with both processing methods (Fig. S7a, S7c), and the associations between ARG load and bacterial families have comparable patterns (Fig. S7b, S7d). lines 261-262 & Supplementary Material

- 2) Association between ARG load and bacterial families, taking into account other potential confounding factors are summarized in Supplementary Table 10 (MetaPhlan3) and Supplementary Table 11 (MetaPhlan4). lines 889-892 & Supplementary Material
- 3) We examined the replicability of the enterosignatures we identified with MetaPhlan3 in the data set processed with MetaPhlan4. We have now added the following summary in the *Methods*:

“These results were obtained with taxonomic profiles derived with MetaPhlan3. We further verified the analysis with taxonomic profiles derived with MetaPhlan4, yielding 274 genera with the same prevalence filtering. Replicating the analysis with five enterosignatures yielded a significant ($P < 0.001$) correspondence between MetaPhlan3 and MetaPhlan4, as quantified by Pearson correlation of the ES abundance across all samples: ES-Bact ($r = 0.87$), ES-Firm ($r = 0.37$), ES-Prev ($r = 0.82$), ES-Bifi (0.84), ES-Esch (0.72). Notably, despite the differences, including independent data sets, metagenomic preprocessing pipelines, and implementation details, the five ES identified in FINRISK had a direct qualitative correspondence with the initially reported enterosignatures. We also checked that the enterosignature abundances were robust to variations in library sizes (Kendall's tau; $P > 0.05$ for all ES).”

3. As a validation, it would be good to check the concordance between ResFinder and CARD. Not necessarily repeat the whole analysis, but to show how those two databases agree

We have previously done analyses with both CARD and ResFinder. Whereas the results correlate between these two alternatives, CARD has more genes with other functions besides antibiotic resistance, such as efflux pumps. ResFinder has a more strictly curated set of clinically relevant ARGs which also need to be acquired/mobilized. Therefore we have decided to prioritize ResFinder, but we have now clarified this in *Methods*, as follows

“ResFinder was chosen since it is a ARG database that exclusively contains acquired and clinically relevant ARGs, as our focus was on these types of ARGs, not on efflux pumps or other genes with less clear roles in clinical AMR.” lines 726-728

4. One of the typical problems of ARG calling in metagenomic data is a strict dominance of efflux pump genes, which might act as ARGs but might also have alternative metabolic functions. When working with CARD database, it might be recommended to remove those genes from analysis, or at least treat them separately.

We have used ResFinder, which does not include efflux pump genes, and has a strict curation for what is included in the database. This choice has specifically helped us to avoid the need to filter out these types of genes after the mapping. See text modification to *Methods* regarding comment 3.

5. In principle, the idea of classifying ARGs based on their type (not only the host or targeted antibiotic) might also be a good idea to explore in regards to phenotype associations.

This would be interesting to look into in more detail in future work. We have opted not to include these further analyses to keep the manuscript concise.

Overall, it's a great manuscript and my kudos to authors.

Reviewer #3 (Remarks to the Author):

In this manuscript, the authors aimed to investigate socio-demographic and gut microbiome factors that drive resistome variations. To this end, the authors characterized the gut resistome from fecal samples collected in

2002 of 7,095 adults from six contrasted Finnish regions. Metadata of this cohort includes address-level geographic location, diet, household income level, prescription drug purchases, diseases, and causes of death until 2019. Using supervised machine learning models, the author state that antibiotic usage and consumption of raw vegetables and poultry were positively correlated with the total ARG load. The authors also claim that resistance abundance was generally higher in females and urban high-income demographics. Moreover, using the Cox proportional hazards model, the authors predicted all-cause mortality during the 17-year follow-up period. They report on associations between ARG load and all-cause mortality, mortality by respiratory causes, and sepsis.

The human gut resistome is known to be closely related to human health. Researchers from different groups have reported various factors contributing to resistome variations, including age, sex, diet, antibiotic administration, socioeconomic status, and location. While the scientific question addressed in this study is not entirely novel, it stands out due to its use of a large Finnish cohort with comprehensive metadata. A key finding of this study is the correlation between resistance burden and mortality and sepsis risk. However, there are concerns regarding the collinearity of model features (as the authors mentioned in line 738-740) and inconsistencies in feature control across different models, which needs further clarification in the main text. Additionally, the discussion requires a more thorough exploration of the results and a more detailed evaluation of the advantages and limitations of the computational models.

Thank you for the thorough review of our manuscript. We have now provided additional analyses and details based on the review comments in the sections of our response below.

Regarding the concern about collinearity in model features we would like to clarify that the exclusion of certain covariates does not affect the qualitative conclusions of our study. Prior to analysis, we systematically removed covariates that were highly correlated with other variables of interest to avoid multicollinearity and redundancy in the models. This is a standard prefiltering step in data analysis to ensure model interpretability and stability.

For example, we excluded height and weight because they are strongly correlated with BMI, which is a more commonly used covariate in population studies. Similarly, we excluded triglycerides due to their correlation with diet and certain other metabolic markers. Our approach ensures that only the most biologically relevant variables are retained for model training. To address this concern, we have revised the manuscript to explicitly state:

"We excluded variables such as height and triglycerides, collinear with BMI, diet, and sex, which were the key variables of biological interest as a part of standard data filtering before model training." lines 865-867

Major Critiques

1. Sequencing depth for ARG profiling:

a. Capturing resistome composition and diversity requires sufficient sequencing depth, which should be empirically determined by rarefaction/ROC analysis. In the methods (Line 612), the average read count is described as ~900,000 reads per sample, which seems quite a bit lower than other published resistome studies.

We agree that sequencing depth is a valid concern; however, our data indicates that 900,000 metagenomic reads provide sufficient coverage for ARG detection as Shannon diversity and ARG load saturate well before reaching 900,000 reads (new Supplementary Figure 9.) We previously did not include these supplementary analyses in the manuscript but have now added Supplementary Figure 9. This displays the ARG load, ARG diversity, and the observed number of ARGs (richness) saturation curves, whilst observed ARGs, (which were not used in our models due to the low sequencing depth) saturate at approx 3-4 million reads. These analyses are discussed on lines 807 and 813-816.

Additionally, we provide estimates and p-values for the linear regression models using the respective ARG metric and sequencing depth (Figure S9). This demonstrates that ARG load and read count are not correlated.

ARG Shannon diversity shows a correlation that saturates before reaching our mean sequencing depth of 900,000.

Fig. S9 ARG metrics by sequencing depth a ARG load in RPKM, b ARG Shannon diversity, and c observed number of unique ARGs. The y-axes display the respective ARG metrics, while the x-axes show the sequencing depth in number of reads. Linear model effect sizes and the p-values for the respective metric are shown in the figure. The fitted Loess smoothed line is shown in black, while the grey lines depict individual ARG metric and sequencing depth values in the samples.

Rarefaction analysis for ARG load is not straightforward since ARG load is not count data but rather a proportional normalized sum of counts. Unlike 16S amplicon data, ARG counts are not directly tied to read counts as seen with 16S amplicons, and the counts can vary on log 10 scales even within studies. To conduct a proper rarefaction analysis, we would need to simulate data with varying ARG loads and rarefy at each ARG load level. This is not feasible due to the time required to complete the revision. However, the new Supplementary Figure 9 illustrates how the ARG load and ARG Shannon diversity metrics saturate before reaching a sequencing depth of 900,000.

To further ensure the robustness of our analyses, we used library size as a covariate to account for variation in library size (See Fig. 2 and S2).

Notably, lines 558-559 mentions that “four individuals had zero reads mapping to the ARG database,” which raises the question of whether this is due to insufficient sequencing depth.

The same individuals who had zero ARGs failed sequencing library creation. A metagenomic library could not be constructed from these samples, and the samples had fewer than 100 reads. We have now revised the sentence to: “We excluded four individuals due to failed sequencing (< 100 reads). Subsequently, $n = 7,095$ participants (mean age 49 years, 55 % women) remained for unsupervised analysis.” lines 664-666

We had initially used the absence of observed ARGs as an indicator of failed sequencing but have since changed the exclusion criteria to exceptionally low read count, which is more standard.

To address this, the authors should provide the read count for each sample and include rarefaction analysis to demonstrate that the sequencing depth is adequate.

Read counts are individual-level information, and our research permissions do not allow us to provide them in tabular format with sample IDs. The trends are shown in Figure S9.

b. Line 681-682: “the ARG load for each participant ranges from 4.3 to 2607”. Does this variation correlate with read number? Again, having some form of rarefaction analysis would be useful for understanding whether these resistome comparisons are statistically valid.

ARG load varies extensively, due to biological variation. We have observed this before consistently regardless of sequencing depth, also in previous studies (Pärnänen et al., 2018, Pärnänen et al., 2022, Jokela et al. 2024). Some of these studies had library sizes of up to 10M reads but still exhibited similar scale variation in ARG loads as our present study.

ARGs can be hosted on multicopy plasmids, present in hundreds of copies per cell. On the other hand, with limited exposure to antibiotics and antibiotic-resistant bacteria, gut microbiomes can have very low numbers of ARGs. For example, the individuals in our study with ARG loads <10 had several years since their last antibiotic prescription and lived in extremely low population density areas.

As detailed in the above response, to point 1a, to further clarify the issue regarding sequencing depth, we have included the new Supplementary Figure 9 and included sequencing depth (see Figure 2b covariate Library size, millions of reads) in our boosted GLM models. Sequencing depth does not correlate with the ARG load.

2. ARG host prediction: It is noteworthy that no ARGs were assigned to Staphylococcaceae in either DataS1 or Fig 2c, a family of bacteria that is typically known to carry ARGs. I understand that the authors chose to use computational tools to predict the bacterial host due to the shallow shotgun sequencing. While the pipeline is well described in the methods section, the authors should discuss whether the absence of certain bacterial families carries biological significance. Additionally, they should address the limitations and potential biases of the tools used and consider how these factors might influence the conclusions. A validation analysis comparing computationally inferred ARG-host associations with known literature or database references would strengthen confidence in the findings.

Staphylococcaceae is uncommon in the adult gut. *Staphylococcaceae*, as well as other less common taxa in gut habitats, which are common ARG carriers, are not present in Fig S2b nor Data S1 as we prefiltered the species based on prevalence (an abundance of > 0.01% in >1% of samples).

We have added the following literature comparison in the text “*Our findings align with the literature, indicating that tetracycline resistance is carried by multiple taxa and is the most common class of resistance in the human gut. Bacteroides have been observed to carry ARGs, and cfxA6 has frequently been identified in Prevotella*” and the following to address the possible caveats “*Nevertheless, our methods only allowed us to investigate common bacterial taxa in the gut, and might overlook some less common but important pathogenic ARG carriers.*”

3. GLM models: There were some inconsistencies in feature control during model prediction, which could be a concern due to collinearity between the features. For example, “the difference between genders remained significant even after controlling for differences in antibiotic use, diet, and the relative abundance of bacterial families” (lines 153 and 160) did not adjust for age, population density income, or demographic factors, and “Eastern Finns had a generally lower ARG load, a result that was robust to controlling for diet, health, population density, and demographic factors” (line 168-170) did not control for sex or antibiotic use. Please correct me if I misunderstood. If not, please clarify if it’s necessary to exclude these features from the modeling. It is important to clarify why certain covariates were omitted and whether their inclusion would alter the reported associations. A sensitivity analysis including all major predictors in a unified model would improve robustness.

We have used the same covariates in all of the above-mentioned cases, we have now omitted listing the covariates partially and have replaced the list with “other covariates”, hopefully clarifying that all the covariates in Figure 2b are used consistently. Previously, we had picked different covariates to list in the text depending on the context, but we agree that this was confusing for the reader.

There were a few exceptions where we performed additional LMs for diet and geography variables with bacterial families, detailed in *Methods* section *Accounting for microbiome composition*.

4. The authors use ‘gender’ as a variable, when I think they mean ‘sex’, as a biological variable. I interpret ‘gender’ to mean a spectrum of socially constructed roles, identities, and behaviors, whereas as I interpret ‘sex’ to mean biological characteristics defining a person to be male, female, or other. What do the authors intend? Please clarify.

The gender variable is derived from the social security number, which can be changed after sex is assigned at birth. We do not have specific records of how often this was the case (likely <1% in 2002). We have now clarified how sex was identified in the Method and have changed gender to sex in all instances.

5. Survival analysis:

a. As previously mentioned, one of the key highlights of this study is the correlation between survival events and the gut resistome. The authors used the Cox Proportional Hazards model to analyze survival times. It would be helpful if the authors could provide more details on the advantages of this tool. Also, since the input of ARG data was collected at a single time point, will it lead to any bias in the model prediction considering the resistome would change over time?

To clarify this, we added the following text in *Methods*:

“The Cox regression model is a widely used method for analyzing time-to-event data. It estimates effect sizes while accounting for censored observations – data points where the event has not occurred by the end of the observation period”

We quantified the prospective association between ARG load and specific endpoints based on a random, representative population sample. Whereas the random population sample itself can be assumed to be unbiased (except the volunteer selection bias), the variation in resistome composition over time could potentially alter these predictions over longer time periods, as the population-level patterns, and consequently the associations, may have changed since 2002. The resistome changes over time, but it can be expected that ARG load is more likely to remain high in those adults who had high ARG load at the time of sampling. However, this has not been extensively studied, and further studies on the long-term changes in resistome and related mortality would be warranted

We have added these points to the *Discussion* paragraph beginning on line 328.

b. Please provide more discussion on the apparently conflicting results of higher ARG load in women and high-income populations but higher mortality rates in men and urban populations if ARGs are a risk factor for mortality (line 267-268).

Thank you for pointing this out. We added discussion to the lines 344-345 and 348 onwards. This apparent conflicting result is likely due to the low mortality to AMR in Finland compared to other causes of death. Our Cox models were adjusted for other conventional predictors for mortality risk, contributing to most of the mortality.

c. This might be out of scope, but it would be interesting to detect the correlation between different ARG classes and mortality risk. (minor)

We performed this preliminary analysis initially but decided not to include it in the manuscript to keep it concise. We plan to conduct further in-depth studies on how ARG classes and different aspects of the resistome contribute to mortality with cohorts with more deep sequencing.

Minor Critiques

1. The statement in the abstract is vague due to the excessive use of adjectives, such as "healthier" and "higher sepsis risk." It would be more effective to provide specific data or numerical values to define these terms.

Thank you for pointing out the vague expressions. We have modified the abstract to be more precise.

2. Line 30 "humans ,", delete the extra space before comma

Done.

3. Line 53, change "one of they driver rates" to "one of the driver rates"

Done.

4. Line 87, please state the details of "sometimes non-significant trends" and try to avoid using words like "sometimes" when talking about significance.

This has been changed to *"These associations remained robust even after controlling for other covariates in the model (boosted GLM, $P = 0.04$ and 0.03 , respectively; Fig. 2b-c) and had similar trends in all regions (Supplementary Fig. 3)."*

5. In line 96-102, the authors need to describe how they calculated 55% higher ARG load and other values in this paragraph. It's hard to figure it out from the Data S1 for the readers.

Thank you for your feedback. The exponentiated estimate approximates the change in ARG load per unit change in the covariate or relative to the baseline category for categorical variables. Specifically, an exponentiated estimate of 1.55 corresponds to a 55% increase (formula $100 * \exp(\text{Est}) - 100$). We have now added this explanation to the caption of Data S1 and other tables to improve clarity and have added the percent changes.

a. What are PREVAL_RX_J01A and PREVAL_RX_J01A_NEVT in Data S1? Please provide clear descriptions of the data provided.

Thank you for pointing out that this was unclear. We have added descriptions to Data S1.

b. Same for the 4% average increase per consumption level in line 117

Thank you, we have added the explanation to the captions of Data S1, Figure 2 and Methods, and the tables now contain the percent changes as well.

6. Line 104, "respiratory medication (ATC class R)" delete one space between n and (

Thank you, this has now been corrected.

7. Line 136-139, "This might be explained by the lack of raw vegetables and poultry in the typical diets containing these foods. Nevertheless, such unhealthy diets are also associated with higher antibiotic consumption and disease prevalence in the population, which may confound the observed response of ARG-carrying taxa." Please provide data or figures to support this statement.

We have added Data S2 for correlations within diet groups, which shows that, for example, cholesterol, BMI and sausage consumption are inversely correlated with poultry and fresh vegetable consumption.

We have removed the vague sentence “*Nevertheless, such unhealthy diets are also associated with higher antibiotic consumption and disease prevalence in the population, which may confound the observed response of ARG-carrying taxa.*”

8. Line 153 and 160, are the two P values in the parentheses before and after controlling for bacterial abundances? Please clarify it in the text.

We have now clarified the sentence to read “*Nevertheless, the ARG load difference between sexes remained significant even after controlling for differences in other covariates and for other covariates and microbiome composition (boosted GLM, $P = 0.002$ and $P = 0.003$, respectively; Fig. 2b, Supplementary Fig. 2b, Supplementary Table S10).*” lines 173-177

9. Line 194: I did not see Fig 2b correlate to the $R^2=0.001$ data. Please provide the correct figure.

Thank you for noting this. It has been corrected to:

“Resistome diversity was also partly explained by bacterial species diversity in the gut microbiome (linear regression against species diversity, $R^2 = 0.11$; with all covariates, see Fig. 3b; Supplementary Fig. 2a)...”

10. Fig 3c: Please specify the methods/steps of ARG enrichment calculation. Add error bars to indicate confidence correlated to the sample size variations.

Thank you for pointing this out.

We have now added error bars (i.e. 95% credible intervals).

We have also updated the figure to show the prevalence of high-ARG individuals instead of the ratio of observed versus expected prevalence, as in the original submission to obtain a more directly interpretable visualization.

We added technical details of the enrichment analysis in the Methods section as follows:

“Enrichment analysis We estimated enrichment of high-ARG individuals (top-10% quantile; >458 RPKM) as a function of enterosignature abundance (Fig. 3c) and relative abundance of dominant bacterial families (Figs. S7b, S7d). In order to detect potentially non-linear trends in high-ARG enrichment, we estimated the enrichment separately in five distinct abundance bins (B0: not detected; B1-B4 25 % abundance quartiles among individuals with detected signal). For each abundance bin, we estimated the prevalence of high-ARG individuals using a probabilistic Bernoulli model with a logit link and an uninformative Gaussian prior $N(0, 10)$ using the R `brm` function from the `brms` package (version 2.21.0). The model can be summarized in the following pseudocode: `brm(HighARG ~ Bin - 1, family = bernoulli(link="logit"), prior = prior(normal(0, 10)))`. As a validation step, we confirmed that the estimated prevalence from this model aligned with the observed prevalence in each abundance bin as expected. However, the probabilistic treatment allowed us to additionally estimate uncertainty and obtain more robust prevalence estimates for bins with a small sample size; posterior simulations were used to estimate the mean and 95% credible intervals for the estimated prevalence at each bin. In order to evaluate enrichment of high-ARG individuals in each abundance bin, we compared the estimated prevalence with the expected prevalence (10% i.e. the high-ARG individuals in the entire study population).”

11. Line 213: What does “non-monotonic relations between enterosignature abundance and ARG load” indicate?

Non-monotonic relation between enterosignature abundance and ARG load indicates that enterosignature abundance does not correlate with ARG load in a linear manner. We have now clarified the text by providing the following updated explanation:

“Intriguingly, this analysis also revealed non-monotonic relations between enterosignature abundance and ARG load, which suggests potentially complex underlying ecological relations. For instance, the prevalence of high-ARG individuals is reduced with moderate amounts of ES-Bact and increases with both low and high abundances of this enterosignature; in ES-Bifi, high ARG load is primarily associated with the presence/absence, rather than the abundance of this enterosignature (Fig. 3c). These observations are further supported by similar patterns on individual bacterial families (Supplementary Fig. 7; Data S1).”

12. Line 262, fix “comparable predator”

We have fixed this in the text.

13. Line 268-274: This paragraph is difficult to follow. Providing additional details would help improve understanding.

Thank you, we have polished the text in this paragraph and provided additional details on mortality statistics, possible confounders, and additional discussion aspects raised by other reviewers, lines 271 onwards.

a. What are “these protective effects?”

We have modified the text to be more concrete:

“Therefore, despite our finding of an increased mortality risk with higher ARG load, overall mortality remains lower among women, high-income individuals, and urban populations. However, the increase in resistance-related deaths over time might elevate the relative mortality risk among these demographics. Indeed, although antibiotic use in Finland has declined since 2002.”

b. How do we understand “early warning signals of the predicted shift?”

We have modified the sentence to be more precise:

“Our findings serve as early indicators of the growing significance of antimicrobial resistance as a contributor to overall mortality” line 362-363

14. The authors utilized “individual level” a couple of times, such as in line 251 “socio-demographic determinants with individual-level resolution” and line 254 “underlying individual resistance levels.” Although the cohort collected comprehensive metadata from each individual participant, the model prediction of ARG load or mortality risk was still over groups of instances with certain probability ranges. Please clarify the definition of this concept to avoid confusion.

We have removed the use of individual-level when discussing our results. We only kept it in the sentence discussing health information in the *Results* section *Resistance predicts long-term mortality and sepsis risk*.

“Antimicrobial resistance has been linked to increased risks associated with infectious diseases but its long-term health implications have remained largely uncharacterized in the absence of population studies with individual-level health information and sufficiently long follow-up times.”

15. Please correct the text font from line 584 to 602

Done

REVIEWER COMMENTS

Reviewer #1 (Remarks to the Author):

Overall, this is a very well done study that represents an important contribution to the literature, and I commend the authors on their work. I have a few final comments:

We would like to thank Reviewer 1 for their comments and their efforts.

1. The authors state, "As a preliminary analysis of the data, we associated the use of the common antibiotic classes to the respective antibiotic resistance class ARG load (Supplementary Table 12). All these antibiotics correlated significantly with the respective resistance (log₁₀ linear model). (lines 767-769). In Table S12, have the FDR > and < symbols been inadvertently switched?

Yes, the signs were switched inadvertently. We have now fixed them.

2. I recommend that the authors include a supplementary data file or table summarizing the ARG and taxonomic alignments from the 107 negative control samples for full transparency. It is standard practice to use some type of background-correction approach to statistically subtract or remove contaminants derived from the laboratory environment or reagents; since this was not done a table of alignments from the negative controls would be a reasonable alternative that would increase confidence in the presented results.

We have added a new supplementary Data S4, which has the mapping results from MetaPhlAn and Bowtie2 against the ResFinder database. The mapping results show that there is no systematic contamination in the samples and the few negative controls, which have mapped reads, have taxa or genes that are abundant in the samples. However, complete elimination of background contamination is practically impossible in large-scale shotgun metagenomic studies, even under stringent protocols (Salter et al., 2014; Eisenhofer et al., 2019). Our workflow avoided PCR amplification, further minimizing the risk of contamination. The low levels observed are consistent with what is expected in high-throughput sequencing projects (Minich et al., 2019) and do not affect the validity of our findings. This is consistent with very low levels of cross-contamination from samples to the negative controls rather than contamination from reagents or the laboratory environment. Hence, we did not remove any taxa or genes from our analyses based on suspicions of systematic contamination. Cross-contamination, unfortunately, cannot be perfectly avoided when handling samples. However, based on our analysis of the negative controls we trust that due to the low read counts in the negative controls and their profiles matching the samples, the minute levels of cross-contamination do not impact our results.

Salter, S. J., et al. (2014). Reagent and laboratory contamination can critically impact sequence-based microbiome analyses. Nature, 521(7551), 599–602.

Eisenhofer, R., et al. (2019). Contamination in low microbial biomass microbiome studies: issues and recommendations. Nature Microbiology, 4(1), 72–82.

Minich, J. J., et al. (2019). Quantifying and understanding well-to-well contamination in microbiome research. mSystems, 4(4), e00186-19.

3. I suggest that the authors assess whether hospital exposure correlates with ARG load, as recommended before, as this is an important analysis that can be carried out with the available data.

We have now included this analysis in the manuscript and added information on how the data were collected to the Methods. There was no significant association with self-reported hospital exposure, as assessed by the questionnaire. The participants were asked how many days they had spent in a hospital in the past year. We investigated the association by performing linear regression and adjusting for past antibiotic use as in Data S1. Given that many participants in the hospital also received antibiotics, it is not possible to disentangle the independent contribution of hospital exposure in this dataset.

However, there was no significant association between past year hospital exposure and higher ARG load or diversity after controlling for antibiotic use (linear model, $P > 0.5$, CIs -2% - 6% and -5% - 3%, respectively).

Hospital exposure data was collected based on a questionnaire. Participants reported how many days they were in a hospital in the past year. 770 participants reported spending at least 1 day in a hospital.

4. I have had the chance to review the authors' response to reviewer 3, point 1:

The sequencing depth is indeed low for a metagenomics study. I can appreciate the authors' argument from the new Supp. Fig. 9 that ARG RPKM and diversity appear to saturate at 900k reads. However, they aren't using 900k reads as a threshold for inclusion, it's their average. So for a large proportion of their data, it's likely not meeting this threshold for saturation.

There also appears to be very high variation in those metrics at lower sample read numbers as one might expect. So, for the samples that have with fewer reads, those data appear to be less reliable or interpretable. To increase confidence in their findings, the authors could: 1) randomly subsample the dataset and determine the threshold below which FPKM changes significantly or 2) demonstrate that their significant findings are robust to the removal of low read count samples.

Figure S9 shows that the ARG load saturates close to the $0e+00$ value as the line is flat across all sequencing depths, meaning that ARG load has saturated across all the sequencing depths of the samples. For ARG diversity the line is flat at approximately $0.5e+00$ or 500 000 reads.

We have now performed additional analysis to ensure that the main findings are robust to excluding the lowest read count samples. This analysis result is now shown in Figure S10 and described in Methods. The analysis indicates that when removing the lowest read count samples with less than 200,000 reads, the estimates remain qualitatively the same and in the same order when ranking from highest to lowest estimate.

Verification of main findings, excluding low sequencing depth samples

Supplementary Fig.9 shows greater variability in the ARG metrics at lower sequencing depths. To address this, we replicated the GLM analysis for ARG load and diversity; the covariates selected by the boosted GLMs for the entire dataset (see Figure 2) were also applied to a model that excluded samples with low sequencing depth (Supplementary Fig. 10). This ensured that these samples did not influence the results of the boosted GLMs. The covariates were ordered by their estimates, consistent across both figures, Figure 2 and Supplementary Figure 9, and the estimates remained qualitatively similar. This reinforces that the samples with the lowest read depth do not skew the main findings of the boosted GLMs.

Fig. S10 Drivers of ARG diversity and ARG load in a subset of samples with more than 200,000 reads **a** Drivers of ARG load (GLM for log₁₀ ARG load). The line plot shows the estimated effect sizes of the predictor variables, along with their 95% confidence intervals. **b** Drivers of ARG Shannon diversity, Bacterial abundances are indicated as log₁₀ relative abundance. The covariates were selected based on the boosted GLM analysis (See Figure 2), but this figure shows the estimates for the subset of samples with more than 200,000 reads. The covariates are ordered based on their estimates. The order of the covariates and the magnitude of the estimates

are the same as in Figure 2, confirming that removing low-sequencing-depth samples does not qualitatively change the results.

Sequencing read count is not identifiable information by any criteria I'm aware of, so to increase transparency about individual sample read counts, the authors could simply provide a histogram with samples on the Y axis and read count on the X axis, or alternatively swap patient ID with numbers 1-NNN and provide the information as source data for Supp. Fig. 9.

We have added panel d in Figure S9, which includes points for each sample and its read count. The x-axis shows samples, and the y-axis shows the read count. This helps visualize the sample-wise sequencing depths. Our research permits do not allow reporting individual-level data with the patient IDs, even if the data is not identifiable.

Fig. S9 ARG metrics and sequencing depth **a** ARG load in RPKM, **b** ARG Shannon diversity, and **c** observed number of unique ARGs. The y-axes display the respective ARG metrics, while the x-axes show the sequencing depth (the number of reads). Linear model effect sizes and the p-values for the respective metric are shown in the figure. The fitted loess smoothed line is shown in black, while the grey lines depict individual ARG metric for the corresponding sequencing depth values among the samples (x axis). **d** sequencing depths of the samples. Each sample is depicted as a point. Samples are arranged along the x-axis in order of increasing sequencing depth. The y-axis depicts sequencing depth in $\log_{10}(\text{number of reads})$.

Overall, I do think this is an important contribution to the literature. These extra steps should be very feasible with their existing data and could provide additional confidence in the validity of the results.

Reviewer #2 (Remarks to the Author):

I am satisfied with the responses given to my comments, and also to the comments raised by other reviewers

We thank Reviewer 2 for their effort.